# Secular change in atmospheric Ar/N$_2$ and its implications for ocean heat uptake and Brewer-Dobson circulation

Shigeyuki Ishidoya[1], Satoshi Sugawara[2,] Yasunori Tohjima[3], Daisuke Goto[4], Kentaro Ishijima[5], Yosuke Niwa[3], Nobuyuki Aoki[1] and Shohei Murayama[1]

[1]National Institute of Advanced Industrial Science and Technology (AIST), Tsukuba 305-8569, Japan
[2]Miyagi University of Education, Sendai 980-0845, Japan
[3]National Institute for Environmental Studies, Tsukuba 305-8506, Japan
[4]National Institute of Polar Research, Tokyo 190-8518, Japan
[5]Meteorological Research Institute, Tsukuba 305-0052, Japan

*Correspondence to*: Shigeyuki Ishidoya (s-ishidoya@aist.go.jp)

**Abstract.** Systematic measurements of the atmospheric Ar/N$_2$ ratio have been made at ground-based stations in Japan and Antarctica since 2012. Clear seasonal cycles of the Ar/N$_2$ ratio with summertime maxima were found at middle to high latitude stations, with seasonal amplitudes increasing with increasing latitude. Eight years of the observed Ar/N$_2$ ratio at Tsukuba (TKB) and Hateruma (HAT), Japan showed interannual variations in phase with the observed variations in the global ocean heat content (OHC). We calculated secularly increasing trends of 0.75±0.30 and 0.89±0.60 per meg yr$^{-1}$ from the Ar/N$_2$ ratio observed at TKB and HAT, respectively, although these trend values are influenced by large interannual variations. In order to examine the possibility of the secular trend in the surface Ar/N$_2$ ratio being modified significantly by the gravitational separation in the stratosphere, 2-dimensional model simulations were carried out by arbitrarily modifying the mass stream function in the model to simulate weakening an enhancement of the Brewer-Dobson circulation (BDC). The secular trend of the Ar/N$_2$ ratio at TKB, corrected for gravitational separation under the assumption of weakening (enhancement) of BDC simulated by the 2D model, was 0.60±0.30 (0.88±0.30) per meg yr$^{-1}$. By using a conversion factor of 3.5x10$^{-23}$ per meg J$^{-1}$ by assuming a 1-box ocean with a temperature of 3.5 °C, average OHC increase rates of 17.1±8.6 ZJ yr$^{-1}$ and 25.1±8.6 ZJ yr$^{-1}$ for the period 2012 – 2019 were estimated from the corrected secular trends of the Ar/N$_2$ ratio for the weakened and enhanced BDC conditions, respectively. Both the OHC increase rates from the uncorrected and weakened-BDC secular trends of the Ar/N$_2$ ratio are consistent with 12.2±1.2 ZJ yr$^{-1}$ reported by ocean temperature measurements, while that from the enhanced-BDC is outside of the range of the uncertainties. Although the effect of the actual atmospheric circulation on the Ar/N$_2$ ratio is still unclear and longer-term observations are needed to reduce uncertainty of the secular trend of the surface Ar/N$_2$ ratio, the analytical results obtained in the present study imply that the surface Ar/N$_2$ ratio is an important tracer for detecting spatiotemporally-integrated changes in OHC and BDC.

# 1 Introduction

The Ar/$N_2$ ratio of air is a unique tracer for detecting changes in the spatiotemporally-integrated air-sea heat flux or ocean heat content (OHC). This is because variations in the Ar/$N_2$ ratio at the Earth's surface are driven by air-sea Ar and $N_2$ fluxes that reflect changes in the solubility of these gases in seawater (e.g. Keeling et al., 2004). The increase in the global OHC is one of the most important parameters for evaluating earth's climate system (e.g. Trenberth and Fasullo, 2010). The relative temperature dependence of the solubility of Ar is larger than that of $N_2$, so that the atmospheric Ar/$N_2$ ratio increases with increasing ocean temperature. Other noble gases behave in a similar way. Bereiter et al. (2018), for example, analyzed Kr/$N_2$ and Xe/$N_2$ ratios in air trapped in ice cores and estimated that the mean global ocean temperature increased by 2.57±0.24 °C over the last glacial transition (20,000 to 10,000 years ago). In terms of ocean heat content, the present-day global OHC increase is evaluated from analysing the ocean temperature measurements using Argo floats (e.g. Cheng et al., 2019). Therefore, precise measurements of the Ar/$N_2$ ratio in air can be used as an independent validation of the OHC estimation from the ocean data. However, long-term changes in the atmospheric Ar/$N_2$ ratio have never been reported so far due to difficulties in detecting a trend with sufficient accuracy, although a few past studies observed its seasonal variations (Keeling et al., 2004; Blaine, 2005; Cassaer et al., 2008; Ishidoya and Murayama, 2014).

We have reported in past studies that the Ar/$N_2$ ratio decreases with increasing altitude in the stratosphere due to gravitational separation of the atmospheric constituents (e.g. Ishidoya et al., 2013; Sugawara et al., 2018). The magnitude of gravitational separation is determined by a balance between mass-independent atmospheric transport, that is, advection and eddy diffusion, and mass-dependent molecular diffusion in the atmosphere. This implies that gravitational separation will be influenced by the atmospheric circulation changes. Therefore, we can use the observed gravitational separation as an indicator of the Brewer-Dobson circulation (BDC) in the stratosphere. There has been no study to evaluate the effect of gravitational separation changes in the stratosphere on the concentrations and isotopic ratios of atmospheric trace gases at the surface, since it has been believed that the gravitational separation signal from the stratosphere is too small to be detected at the surface. Therefore, in our previous studies, we simulated "δ", which is an indicator of gravitational separation derived from the Ar/$N_2$ ratio and stable isotopic ratios of $N_2$, $O_2$ and Ar, using atmospheric transport models by assuming the surface δ to be zero (Ishidoya et al., 2013; Sugawara et al., 2018; Belikov et al., 2019). However, because long-term changes in the Ar/$N_2$ ratio near the surface are expected to be extremely small, e.g. 0.00026 % corresponding to a heat input of 100 ZJ (1 zetajoule = $10^{21}$ J) into a 10 °C ocean (Keeling et al., 2004), a very small secular change in the stratospheric gravitational separation signal near the surface may modify the long-term change in the surface Ar/$N_2$ ratio. If so, an evaluation of gravitational separation of the whole atmosphere is needed for a precise estimate of the global OHC increase based on long-term changes in the surface atmospheric Ar/$N_2$ ratio.

In this paper, we present results from an analysis of 8-year-long measurements of the atmospheric Ar/$N_2$ ratio at the ground surface stations and propose "$\delta_\Omega$" as a new indicator of gravitational separation of the whole atmosphere by using a 2-D model.

By using the simulated $\delta_\Omega$, we derived the secular trend of the Ar/N$_2$ ratio corrected for gravitational separation changes associated with the BDC change. Finally, we estimated the global OHC change based on the corrected Ar/N$_2$ ratio.

## 65  2 Experimental procedures

Atmospheric Ar/N$_2$ has been observed at Tsukuba (TKB; 36°N, 140°E), Japan continuously since February 2012 (Ishidoya and Murayama, 2014). Located on the roof of a laboratory building of the National Institute of Advanced Industrial Science and Technology (AIST), sample air is taken from an air intake by using a diaphragm pump with gas velocity higher than 5 ms$^{-1}$ (4 mm ID and a flow rate of 4 L min$^{-1}$) at the tip of the air intake to prevent thermally-diffusive inlet fractionation (Sturm et

70  al., 2006; Blaine et al., 2006). The sample air is introduced into a 1 L stainless steel buffer tank after water vapour in the air is reduced by using an electric cooling unit at 2°C. The gas is then exhausted from the buffer tank at a flow rate of about 4 L min$^{-1}$. A small portion of this exhausted gas is introduced into a 1/8-inch OD stainless-steel tube and any remaining water vapour is removed using a cold trap at -90 °C. Finally, the remaining sample air is vented through an outlet path at a rate of about 10 mL min$^{-1}$, and only a miniscule amount of it is transferred to the ion source (or waste line) of a mass spectrometer (Thermo

75  Scientific Delta-V) through an insulated thin fused-silica capillary. As for the reference air, it is always supplied from a high-pressure cylinder at a flow rate of about 4 mL min$^{-1}$, and a miniscule amount of it is introduced into the ion source (or waste line) of the mass spectrometer through another fused-silica capillary. For the continuous measurements, alternate analyses of the sample and reference air are made repeatedly. The measurement time required to obtain one data value is 62 seconds, but we usually use the 550-data averaged value as the reported Ar/N$_2$ ratio obtained from the continuous observations (about 11-

80  hour averaged value). We also measure stable isotopic ratios of N$_2$, O$_2$ and Ar, and O$_2$/N$_2$ ratio and CO$_2$ mole fraction simultaneously, and use the 550-data averaged values for the stable isotopic ratios and one data value without averaging for O$_2$/N$_2$ ratio and CO$_2$ mole fraction; these measurements constitute the continuous observations. Details of the continuous measurement system used are given in Ishidoya and Murayama (2014).

We have also collected air samples at a rate of once per month at Hateruma Island (HAT; 24°N, 124°E) and Cape Ochiishi

85  (COI; 43°N, 146°E), Japan since July 2012 and October 2013, respectively, and at Syowa station (SYO; 69°S, 40°E) since January 2016, for the analyses of the atmospheric Ar/N$_2$ ratio. Each air sample is taken from an air intake by using a diaphragm pump at a flow rate of about 5 L min$^{-1}$ and filled into a 1 L stainless steel flask whose inner walls are silica-coated after removing water vapour using a cold trap (-40 °C at HAT and COI, and -80 °C at SYO). Similar to our previous study (Ishidoya et al., 2016), the 1 L stainless steel flasks are equipped with two metal-seal valves on each side to equalize the inner pressure

90  to the pressure between the two metal-seal valves to prevent a mass-dependent fractionation due to small leak through the valve. During air sampling, the inner pressure of the flask is kept at an absolute pressure of 0.2 MPa using a backpressure valve (Tohjima et al., 2003). The sample air collected in the flasks are then sent to AIST and analyzed by using the same mass spectrometer as described above. The sample air is supplied from the flask at a flow rate of about 4 mL min$^{-1}$ through a cold trap (about -50 °C), and a miniscule amount of it is introduced into the ion source (or waste line) of the mass spectrometer

through a fused-silica capillary. It is noted that we also used 760 mL glass flasks with a Viton O-ring seal valves on each side for collecting air samples at HAT and COI prior to September, 2015 and January, 2019, respectively. However, we found slight seasonally dependent differences in the $Ar/N_2$ and $O_2/N_2$ ratios between the analytical results from the 1 L stainless steel flasks and those from the 760 mL glass flasks. It is interesting to note that the $O_2/N_2$ ratio from the 1 L stainless steel flasks agree well with the $O_2/N_2$ ratio reported by the National Institute for Environmental Studies (NIES) (e.g. Tohjima et al., 2019), considering the difference in the span sensitivity of the $O_2/N_2$ ratio between the AIST and NIES (our unpublished data). Therefore, we have decided to adopt the $Ar/N_2$ data obtained from the 1 L stainless steel flasks and correct the data from the 760 mL glass flasks based on the comparison of the $Ar/N_2$ ratios measured from the stainless steel flasks and the glass flasks at HAT for the period October, 2015 – January, 2019. Cause(s) of such an offset between the stainless steel flasks and glass flasks have not been determined yet, but it may be related to the seasonal difference in the ambient temperature during the time the flasks were shipped from the observational sites to our laboratory and its effect on the condition of Viton O-ring seal valves used in the glass flasks.

The $Ar/N_2$ ratio is usually reported in per meg units as follows.

$$\delta(Ar/N_2) = \left( \frac{\left( [n_{Ar}] \middle/ [n_{N_2}] \right)_{sample}}{\left( [n_{Ar}] \middle/ [n_{N_2}] \right)_{standard}} - 1 \right) \times 10^6. \qquad (1)$$

Here, the subscripts 'sample' and 'standard' indicate the sample air and the standard gas, respectively. Because Ar constitutes 9,334 $\mu$mol mol$^{-1}$ of air by volume (Aoki et al., 2019), 5 per meg of $\delta(Ar/N_2)$ is equivalent to about 0.05 $\mu$mol mol$^{-1}$. In this study, $\delta(Ar/N_2)$ of each air sample was determined against our primary standard air (cylinder No. CRC00045) using the mass spectrometer Thermo Scientific Delta-V. Our air standards, which are classified into primary and secondary, are dried ambient air or industrially-purified air-based $CO_2$ standard filled in 48-L high-pressure cylinders. As shown in Fig. 1, variations in the annual average $\delta(Ar/N_2)$ of our 3 secondary standards are within $\pm0.9 - \pm2.2$ per meg ($\pm1.6$ per meg, on average) and nearly stable for 8 years with respect to the primary standard. Therefore, we allowed an uncertainty of $\pm1.6$ per meg for the annual average $\delta(Ar/N_2)$ observed in the present study associated with the stability of the standard air, which corresponds to an uncertainty of $\pm0.28$ per meg yr$^{-1}$ for the 8-year-long secular trend. We have also prepared high-precision gravimetric standard mixtures of Ar, $O_2$, $N_2$ and $CO_2$, with standard uncertainties for the Ar and $O_2$ molar fractions of 0.6 to 0.8 $\mu$mol mol$^{-1}$ (Aoki et al., 2019). From the measurements of the gravimetrically-prepared standard mixtures by the mass spectrometer, it was confirmed that the span sensitivity of $\delta(Ar/N_2)$ obtained from the mass spectrometer agreed to within 0.2 % of the gravimetric values of the standard mixtures, in the range from -4,500 to +1,800 per meg of $\delta(Ar/N_2)$ against our primary standard air.

In the present study, we use 1-week averaged values from the continuous observation at Tsukuba after implementing the following data selection procedure. First, the $\delta(Ar/N_2)$ values with $\delta^{15}N$ higher than 3.0 per meg were excluded from the analyses. As mentioned above, the $\delta(Ar/N_2)$ and $\delta^{15}N$ values are already 11-hour averaged values. We have found these high

$\delta^{15}N$ events accompanied by high $CO_2$ events, as well as occasionally with slightly high $\delta(Ar/N_2)$ events, especially in the winter, but they did not correlate with variations in the isotopic ratio of $O_2$ and Ar. Therefore, it appears that some unspecified interferences of mass 29 (possibly $^{13}C^{16}O$ and/or fragments of hydrocarbons) rather than the molecular-diffusive separation of $^{15}N^{14}N$ and $^{14}N^{14}N$ must have been superimposed on the observational results of $\delta^{15}N$ during these events. It is possible that these high $CO_2$ and $\delta^{15}N$ events occur under a stable atmospheric condition in the winter, so that simultaneously observed

$\delta(Ar/N_2)$ may also be modified by local effects such as thermally-diffusive separation of Ar and $N_2$ due to a temperature inversion near the surface (Adachi et al., 2006). Therefore, the threshold value of 3.0 per meg was determined to be reasonable, considering that an average $\delta^{15}N$ value of $1.1\pm1.7$ per meg was observed at HAT for the period October 2015 – January 2020, while the observed $CO_2$ mole fractions over the same period were much closer to the values of the background air than those observed at TKB. After the above data selection procedure, 1-week averaged values of $\delta(Ar/N_2)$ were calculated. The

measurement uncertainty of the 1-week averaged values of the continuous observation was estimated to be about $\pm3$ per meg as a standard deviation from the best-fit curve shown in Fig. 2 discussed below, while that of the flask air sample measurements was estimated to be about $\pm7$ per meg from repeated analyses of the same air samples.

## 3 Results and discussion

### 3.1 Atmospheric $\delta(Ar/N_2)$ observed at ground surface stations in Japan and Antarctica

Figure 2 shows atmospheric $\delta(Ar/N_2)$ observed at COI, TKB, HAT and SYO. Best-fit curves to the data and long-term trends obtained using a digital filtering technique (Nakazawa et al., 1997) are also shown. Using this filtering technique, the average seasonal cycle of $\delta(Ar/N_2)$ was modelled by a fundamental sine-cosine and its first harmonic with the respective periods of 12 and 6 months. The residuals obtained by subtracting the average seasonal cycle from the data were interpolated linearly to calculate the daily values of $\delta(Ar/N_2)$, which were then smoothed by a 26th-order Butterworth filter with a cutoff period of 36

months to obtain the interannual variation. The interannual variation was then subtracted from the original data, and the average seasonal cycle was determined again from the residuals. These steps were repeated until we obtained an unchanging interannual variation. In this study, we have defined an average linear increasing/decreasing trend as the "secular trend" (as in Fig. 5 discussed below).

As seen in Fig. 2, we can distinguish clear seasonal $\delta(Ar/N_2)$ cycles at COI, TKB and SYO and some interannual variations at

TKB and HAT. Figure 3 shows the detrended values of $\delta(Ar/N_2)$ and average seasonal $\delta(Ar/N_2)$ cycles observed at all 4 sites. The seasonal maxima were found in the summertime, which is expected since the sea surface temperatures around the observational sites reach seasonal maximum in the summertime, due to the larger relative temperature dependent solubility of Ar compared to that of $N_2$. The peak-to-peak amplitudes of the seasonal $\delta(Ar/N_2)$ cycles were $21\pm10$, $11\pm4$, $5\pm10$ and $32\pm9$ per meg at COI, TKB, HAT and SYO, respectively. The uncertainties for the amplitudes indicate standard deviations of the

detrended values from the average seasonal cycle. The uncertainties at COI, HAT and SYO were found to be not only larger

than that at TKB, but also ±7 per meg of the uncertainty originated from the repeated analyses of the same flask air sample. This would be due to the fact that the uncertainty of each analysis value of the standard air (about ±5 per meg, black dots in Fig. 1) is superimposed on the uncertainty of each analysis of the flask air sample and continuous measurement. Therefore, a mean squared error expected for the observational data from the flask air sample is about ±9 per meg, which is comparable to

the uncertainties in the seasonal amplitudes in Fig. 3. Corrections to the $\delta(Ar/N_2)$ data at HAT and COI prior to September, 2015 and January, 2019 discussed above (corrections based on the comparison of the $\delta(Ar/N_2)$ values from the stainless steel flasks and the glass flasks), could also be contributing to the uncertainties.

Similar increases in the seasonal $\delta(Ar/N_2)$ cycle amplitude with increasing latitude were also observed by Keeling et al. (2004) and Cassar et al. (2008). Cassar et al. (2008) also reported on the seasonal $\delta(Ar/N_2)$ cycle at SYO with a peak-to-peak amplitude

of 21±8 per meg. The amplitude at SYO found in this study is consistent with that found by Cassar et al. (2008), within the quoted uncertainties. For La Jolla (LJO; 33°N, 117°W), USA located at a similar latitude as TKB, Keeling et al. (2004) and Blaine (2005) found a seasonal $\delta(Ar/N_2)$ cycle with a peak-to-peak amplitude of about 10 per meg. This agrees with the amplitude at TKB observed in this study. Seasonal minima and maxima at SYO and LJO reported by Cassar et al. (2008) and Keeling et al. (2004), respectively, are in general agreement with those observed at SYO and TKB in this study. On the other

hand, the seasonal $\delta(Ar/N_2)$ cycle at HAT was not so clear but the average peak-to-peak amplitude may be slightly smaller than the 14±6 per meg observed at Kumukahi (20°N, 155°W), USA located at a similar latitude to HAT (Keeling et al., 2004). Similar and consistent results obtained by this and other studies give confidence in our ability to capture natural variations of $\delta(Ar/N_2)$ in the atmosphere.

Figure 4 shows variations in $\delta(Ar/N_2)$ observed at COI, TKB and HAT, after subtracting the seasonal cycles and shorter-term

variations of less than 36 months. For analyses of the interannual variation, we mainly used the data from TKB and HAT since the observations at these stations date back to 2012 and are longer than data from other sites. Variations in the 0-2000 m global OHC are shown (Fig. 4), reported by the National Oceanographic Data Center (NOAA)/National Centers for Environmental Information (NCEI) (updated from Levitus et al. 2012, https://www.nodc.noaa.gov/OC5/3M_HEAT_CONTENT/). The OHC values are shown as anomalies from the baseline value observed in mid-1980s. In the figure we also plot interannual variation

of the OHC values obtained by using the same digital filtering technique used in Fig. 2, and globally averaged surface temperature anomalies (Japan Meteorological Agency, http://www.data.jma.go.jp/cpdinfo/temp/nov_wld.html). Rates of change of the interannual variations of $\delta(Ar/N_2)$ and OHC are also shown by the red lines. As can be seen from the figure, $\delta(Ar/N_2)$ show significant interannual variations and slight secular increasing trends while the global OHC show a more prominent secular increase. Moreover, the observed variations in the change rates of $\delta(Ar/N_2)$ and OHC are quite similar to

each other in phase, suggesting a strong correlation between the large-area air-sea heat flux and the interannual variations in $\delta(Ar/N_2)$. The minima of the rate of change of $\delta(Ar/N_2)$ and OHC appeared at the beginning of 2016 when a maximum in the surface temperature anomaly appeared. This correspondence is qualitatively reasonable since a decrease in the OHC change rate indicates a decrease in the net ocean heat uptake, and thus leading to an increase in surface temperature.

As mentioned above, the ratio of interannual variation to secular increase is larger for $\delta(Ar/N_2)$ than for OHC. To examine this difference, we estimated the interannual variation of the atmospheric $\delta(Ar/N_2)$ expected from the air-sea Ar and $N_2$ fluxes caused by the interannual variation in the global OHC. We converted the change rate of OHC to that of $\delta(Ar/N_2)$ by assuming a coefficient of $3.5 \times 10^{-23}$ or $3.0 \times 10^{-23}$ per meg $J^{-1}$, which was derived from the following equations (Keeling et al., 1993).

$$F_{Ar} = -\frac{dC_{eq_{Ar}}}{dT}\frac{\dot{Q}}{C_P}, \qquad (3)$$

$$F_{N_2} = -\frac{dC_{eq_{N_2}}}{dT}\frac{\dot{Q}}{C_P}. \qquad (4)$$

Here, $F_{Ar}$ ($F_{N2}$) is the net sea-to-air Ar ($N_2$) flux in moles $m^{-2}\ s^{-1}$, $dC_{eq\_Ar}$ ($dC_{eq\_N2}$) is the temperature derivatives of the solubility of Ar ($N_2$) in mole $m^{-3}\ K^{-1}$ (Weiss, 1970), $\dot{Q}$ is the net air-to-sea heat flux in J $m^{-2}$, and $C_P$ is the heat capacity of seawater in J $m^{-3}\ K^{-1}$. In eqs. (3) and (4), the assumption that surface waters fully equilibrate could lead to overestimation of the gas fluxes. As a first approximation we modeled the global ocean as 1-box and assumed an average temperature of 3.5 or 7.5 °C. The temperatures of 3.5 and 7.5 °C, corresponding to the respective coefficients of $3.5 \times 10^{-23}$ (including deep water) and $3.0 \times 10^{-23}$ per meg $J^{-1}$ (not including deep water), were the average values of the ocean model shown in Fig. 1 of Bereiter et al. (2018) for the present-day ocean. We also assumed constant $C_P$ and salinity of $3.9 \times 10^6$ J $m^{-3}\ K^{-1}$ and 35 ‰, respectively. To convert $F_{Ar}$ and $F_{N2}$ to changes in atmospheric $\delta(Ar/N_2)$, we employed $5.124 \times 10^{21}$ g for the total atmospheric mass of dry air (Trenberth, 1981), 28.97 g $mol^{-1}$ for the mean molecular weight of dry air, $3.6 \times 10^{14}$ $m^2$ for the surface ocean area of the earth, and respective fractions of 0.00933 and 0.7808 for Ar and $N_2$ in the atmosphere.

As a result, the interannual variation in the change rates of the global OHC was estimated to drive only 10 % of that of the atmospheric $\delta(Ar/N_2)$. This discrepancy would be due to the fact that the troposphere, and not only the ocean, does not mix perfectly on a timescale of a year, and that the surface-ocean temperature anomalies would be a large source of interannual variation on a yearly timescale for the observed $\delta(Ar/N_2)$ in the near-surface air. Therefore, in order to interpret the relationship between the interannual variations of OHC and $\delta(Ar/N_2)$ more quantitatively, we need to consider the local heat anomalies and atmospheric transport effects. In other words, a much longer-term $\delta(Ar/N_2)$ variation is less affected by the surface-ocean anomalies and atmospheric transport, but will include influence signature of the average temperature from the surface to the deeper part of the ocean. In this regard, Suga et al. (2008) estimated that the renewal time of permanent pycnocline water in the North Pacific to be 2 – 4 years for eastern subtropical mode water, 2 and 5 – 9 years for the lighter and denser subtropical mode water, respectively, and 10 – 20, 20 – 30 and >60 years for the lighter, middle and denser central mode water, respectively. Therefore, it is expected that the secular trend of the $\delta(Ar/N_2)$ values obtained during 2012 – 2019 (8-years) in the present study does not reflect the deep OHC change.

Figure 5 shows the $\delta(Ar/N_2)$ values observed at TKB and HAT, along with their best-fit curves and interannual variations from Fig. 2. In the figure, we have also plotted the annual average values of $\delta(Ar/N_2)$ obtained at these sites. The data are expressed as anomalies from the average value for the observation period. From the figure, we can distinguish secularly increasing trends of $\delta(Ar/N_2)$ at both of the sites throughout the observation periods, with large interannual variations superimposed on these

trends. Therefore, we applied another best-fit curve consisting of the fundamental and its first harmonics and a linear secular trend to the observed data (black dotted lines), to extract the secular trends as a component of the linear trend of the best-fit curve (thick black solid lines). The secular trends were found to be 0.75±0.30 and 0.89±0.60 per meg yr$^{-1}$ at TKB and HAT, respectively. The uncertainties for TKB (HAT) were evaluated, taking into account the respective uncertainties of ±0.11 and ±0.28 per meg yr$^{-1}$ (±0.53 and ±0.29 per meg yr$^{-1}$) for the regression and the long-term stability of the standard air. Comparable secular trends of 0.70±0.33 per meg yr$^{-1}$ for TKB and 1.06±0.61 per meg yr$^{-1}$ for HAT were also obtained from the difference in the annual average value between 2012 and 2019 for each station. Although it is possible these changes observed over the study period might not represent a long-term trend, but is part of a large interannual variation. Nevertheless, it would of interest to see if it is possible to obtain a scientifically "meaningful" OHC change based on the secular $\delta(Ar/N_2)$ trend reported earlier. Accordingly, we used the secular trend at TKB (0.75±0.30 per meg yr$^{-1}$), because of the smaller uncertainty than that obtained at HAT, to propose a method to estimate the global OHC or BDC change based on the surface and the stratospheric secular trends of $\delta(Ar/N_2)$. For this purpose, we simulate gravitational separation effect on the $\delta(Ar/N_2)$ in the whole atmosphere by using a 2-D model described below in subsection 3.2.

### 3.2 Simulation of gravitational separation and its effect on $\delta(Ar/N_2)$ at ground surface

As mentioned in the Introduction, we have reported observational results of gravitational separation in the stratosphere (Ishidoya et al., 2008, 2013, 2018; Sugawara et al., 2018). Those results showed that stratospheric $\delta(Ar/N_2)$ also decreases rapidly with increasing altitude above the tropopause. Also, not only are there large year-to-year variations in the difference between the stratospheric and tropospheric $\delta(Ar/N_2)$ values, year-to-year variations in stratospheric $\delta(Ar/N_2)$ are much larger than those observed in the troposphere (Ishidoya et al., 2013; 2018). Therefore, we need to explore the possibility of tropospheric $\delta(Ar/N_2)$ variations caused by changes in the stratospheric gravitational separation that influence the whole troposphere. We have compared the observed and simulated gravitational separation of atmospheric major components above the tropopause in previous studies (Ishidoya et al., 2013, 2018; Sugawara et al., 2018; Belikov et al., 2019), but we did not consider gravitational separation in the troposphere. Therefore, in this study we updated the SOCRATES model (Huang et al., 1998) to calculate variations in Ar and $N_2$ from the surface to 120 km, taking into account molecular diffusion processes generating gravitational separation. The SOCRATES is a 2-D model with parameterized eddy diffusivity coefficients, tuned to produce a realistic stratospheric age of air distribution. In the present study we carry out sensitivity studies with arbitrarily modifying the mass stream function in the model. Here, we describe only those modifications we made to the model for the calculation of $\delta(Ar/N_2)$. Additional description of the SOCRATES model is presented in Appendix A.

First, our 2-D model was expanded to be able to calculate explicitly any inert gas components and their isotopic ratios and elemental ratios, including $\delta(Ar/N_2)$. In our previous model studies, we used $^{44}CO_2$ and $^{45}CO_2$ to reproduce gravitational separation for the sake of simplicity. However, it is now necessary to explicitly include the molecular diffusion coefficients of

all gas components in order to reproduce the molecular diffusion processes in the stratosphere more accurately, because the molecular diffusion coefficient is dependent on the molecular mass. In this study, the molecular diffusion coefficient of the gas component A in air ($D_A$) was calculated using the following equation (Poling, et al., 2001).

$$D_A = \frac{1.43\times10^{-4}T^{1.75}}{p\sqrt{m_{A-air}}\left(\sqrt[3]{\sigma_A}+\sqrt[3]{\sigma_{air}}\right)^2} \quad (5)$$

Here, $T$ and $p$ are the temperature (K) and pressure (hPa), respectively. The reciprocal average of $m_A$ and $m_{air}$, $m_{A-air}$, is defined as follows.

$$\frac{1}{m_{A-air}} = \frac{1}{2}\left(\frac{1}{m_A} + \frac{1}{m_{air}}\right) \quad (6)$$

Here, $m_A$ and $m_{air}$ are the molecular masses of component A and air, respectively. $\sigma_A$ and $\sigma_{air}$ are the diffusion volumes of the component A and air, respectively. The diffusion volumes of $^{40}Ar$, $^{28}N_2$, and air are 16.2, 18.5, and 19.7, respectively (Poling et al, 2001).

Second, a new δ value has been defined and used for δ(Ar/N$_2$) in this study. Usually, the isotopic ratios or elemental ratios of the atmospheric major compositions are expressed as values relative to their ratios in the troposphere. Therefore, it is common to treat the δ values at the ground surface as zero. However, high-precision analyses have recently revealed that, as is the case in δ(Ar/N$_2$), the δ values at the ground surface are not always constant, and they have seasonal and inter-annual variations, as well as long-term trends (Keeling et al., 2004; Cassar et al., 2008; Blaine, 2005; Bent, 2014). The purpose of our analysis here is to evaluate the effect of gravitational separation on the δ value at the ground surface using the 2-D model. Therefore, it is no longer appropriate to assume that the δ value at the ground surface is always zero. Observationally, small variations at the ground surface can be detected by using specific gas cylinders as constant references. In a numerical model, a constant reference is also needed for evaluating the effects of gravitational separation on the δ value at the ground surface. In our 2-D model, we used the ratio of total amount (M) of each gas component in the model atmosphere as the constant reference and defined a new δ value, $\delta_\Omega$, by the following equation.

$$\delta(Ar/N_2)_\Omega = \left(\frac{[n_{Ar}]/[n_{N_2}]}{M_{Ar}/M_{N_2}} - 1\right) \times 10^6 \quad (7)$$

The total amount was calculated by integrating from the ground surface to the altitude of 120 km. According to this definition, $\delta_\Omega$ will be zero if there are neither molecular separations nor sinks/sources in the entire atmosphere. In the actual atmosphere, $\delta_\Omega$ becomes negative in the stratosphere due to gravitational separation, but will be a small positive value in the troposphere. A 50-year-long spin-up calculation was carried out for δ(Ar/N$_2$)$_\Omega$ under steady-state condition, and we found that δ(Ar/N$_2$)$_\Omega$ in the troposphere reached a steady-state value of about 30 per meg. In other words, the tropospheric Ar/N$_2$ ratio is enriched by about 30 per meg relative to the homogenous atmosphere due to the atmospheric gravitational separation. If the gravitational

separation is strengthened due to atmospheric circulation changes in the stratosphere, $\delta_\Omega$ at the ground surface will slightly increase. On the other hand, if the gravitational separation is weakened in the stratosphere, $\delta_\Omega$ at the ground surface will decrease. Therefore, by using $\delta_\Omega$, it is possible to examine how a change in the gravitational separation in the stratosphere affects $\delta_\Omega$ at the ground surface. Meridional distribution of $\delta(Ar/N_2)_\Omega$ calculated using the updated SOCRATES model, and its

comparison with the stratospheric $\delta(Ar/N_2)$ observed in our previous studies, are presented in Appendix B.

Third, age of air (AoA) is calculated using an idealised tracer. In our previous model calculations, the AoA was calculated from the $CO_2$ mole fraction. However, since the actual increase in $CO_2$ mole fraction given at the ground surface is not a linear increase but includes non-linear trends and inter-annual fluctuations, it is not possible to estimate the correct AoA from the simple lag time method (e.g. Waugh and Hall, 2002). Therefore, in this study, we introduced an inert idealised tracer that

increases linearly at the ground surface, and calculated the AoA in the stratosphere from the mole fraction of this idealised tracer.

### 3.3 Secular trends in the observed and simulated $\delta(Ar/N_2)$ and its implication for changes in OHC and BDC

To examine how $\delta(Ar/N_2)_\Omega$ is influenced by changes in the BDC, model simulations were made by arbitrarily changing the mean meridional circulation (MMC) represented by mass stream function in the 2-D model so that AoA changed by $\pm 0.02$ yrs

yr$^{-1}$ at 35 km over the northern mid-latitudes. The simulated AoA values for the 35-km height are shown in Fig. 6. Here the positive and negative rates correspond to the weakening and enhancement of the BDC simulations, respectively. These simulations with fixed horizontal mixing are not enough to represent the mechanism to drive the observed AoA change in the real atmosphere, but it does serve as a kind of sensitivity test. In this regard, we present the annual mean meridional distribution of the AoA trend for the weakened and enhanced BDC simulations in Appendix B. Based on some past studies, we have

adopted a value of $\pm 0.02$ yrs yr$^{-1}$ as AoA trends for the BDC simulations : for example, Diallo et al (2012) reported an increase in AoA of about 0.03 yrs yr$^{-1}$ in the middle stratosphere for the period 1989-2010 based on the ERA-Interim reanalysis data, while Fritsch et al. (2016) reported AoA unchanged within uncertainties over the northern mid-latitudes for the period 1975-2016 based on the observations of $CO_2$ and $SF_6$ updated from Engel et al. (2009), and Waugh (2009) showed, based on results from chemistry–climate models, a negative AoA trend between -0.005 and -0.02 yrs yr$^{-1}$ for the time period and location

considered by Engel et al. (2009). The secular AoA trends forced in our study fall within the range of these studies, although the observation periods do not overlap with each other. We regard these simulations to be a first step in investigating the effect of gravitational separation of the whole atmosphere on $\delta(Ar/N_2)$ at the surface.

Secular changes in the simulated $\delta(Ar/N_2)_\Omega$ at the surface obtained from the weakened and enhanced BDC simulation, and those at 35 km are also shown in Fig. 6. As can be seen from the figure, clear inverse relationships are found between the

secular trends of $\delta(Ar/N_2)_\Omega$ at the surface and those at 35 km. In the weakened BDC simulation, $\delta(Ar/N_2)_\Omega$ changes secularly by 0.15 and -4.5 per meg yr$^{-1}$, respectively, at the surface and 35 km. In the enhanced BDC simulation, respective secular changes in $\delta(Ar/N_2)_\Omega$ by -0.13 and 4.0 per meg yr$^{-1}$ at the surface and 35 km are found. As discussed above, atmospheric

$\delta(Ar/N_2)$ is estimated to increase by $3.5 \times 10^{-23}$ per meg when 1 J of heat is added to a 3.5 °C ocean water mass. Based on this relationship, when OHC increases by 93 ZJ during 2012 – 2019, which is the global 0 – 2000 m OHC increase from the

NOAA/NCEI data, the increase rate in atmospheric $\delta(Ar/N_2)$ is estimated to be 0.46 per meg yr$^{-1}$. Therefore, 0.15 (-0.13) per meg yr$^{-1}$ for the surface $\delta(Ar/N_2)_\Omega$ due to the weakening (enhancement) of the BDC is a non-negligible trend compared with the expected changes in atmospheric $\delta(Ar/N_2)$ due to the global OHC increase.

Figure 7 shows the same annual average values and secular trend of atmospheric $\delta(Ar/N_2)$ at TKB as in Fig. 5. We calculated an increase rate of $\delta(Ar/N_2)_{cor}$ by subtracting 0.15 per meg yr$^{-1}$ from the observed secular trend of 0.75±0.30 per meg yr$^{-1}$ to

remove the estimated effects of $\delta(Ar/N_2)_\Omega$ increase obtained from the weakened BDC simulation. Thus, the derived secular increase rate of $\delta(Ar/N_2)_{cor}$, which represents the increase rate of $\delta(Ar/N_2)$ driven only by the OHC change, is 0.60±0.30 per meg yr$^{-1}$. We can convert this increase rate to the global OHC increase rate by assuming above-mentioned coefficient of 3.5 (3.0) $\times 10^{-23}$ per meg J$^{-1}$ (hereafter referred to as the increase rate of "OHC$_{ArN2}$"). The obtained secular increase of OHC$_{ArN2}$ by 17.1±8.6 (20.2±10) ZJ yr$^{-1}$ is shown in Fig. 7, together with that of the global 0 – 2000 m OHC reported by NOAA/NCEI

(hereafter referred to as "OHC$_{oc}$"). The increase rate of OHC$_{ArN2}$ is consistent with the OHC$_{oc}$ value of 12.2±1.2 ZJ yr$^{-1}$ within uncertainties. Recently, by extracting a solubility-driven component of APO from their atmospheric O$_2$/N$_2$ ratio and CO$_2$ measurements and ocean model simulations, Resplandy et al. (2019) estimated an increase rate of the global OHC to be 12.9±7.9 ZJ yr$^{-1}$ for the period 1991 – 2016 that is consistent, within the uncertainties, with the OHC$_{oc}$ during the same period. The results of the present study and Resplandy et al. (2019) suggest the usefulness of atmospheric observations for independent

confirmation of ocean heat uptake estimated from ocean temperature measurements.

As described above, the increase rate in OHC$_{ArN2}$ obtained by using $\delta(Ar/N_2)_\Omega$ from the weakening BDC simulation agrees with the OHC$_{oc}$ estimate. On the other hand, the increase rate in OHC$_{ArN2}$ of 25.1±8.6 (29.6±10) ZJ yr$^{-1}$, obtained by using $\delta(Ar/N_2)_\Omega$ from enhanced BDC simulation assuming the coefficient of 3.5 (3.0) $\times 10^{-23}$ per meg J$^{-1}$, is significantly larger than the OHC$_{oc}$ estimate. In general, modeling studies have pointed to an accelerating BDC due to anthropogenic climate change

(e.g. Austin and Li, 2006). However, the increase rate in OHC$_{oc}$ agrees well OHC$_{ArN2}$ based on the weakened BDC simulation rather than on the enhanced BDC simulation. In this regard, several balloon-borne observations (Ray et al., 2014) and the ERA-Interim reanalysis data (Diallo et al., 2012) have suggested aging of air in the northern hemispheric midlatitude mid-stratosphere, implying a slowdown in the deep northern hemispheric branch of the BDC. Garfinkel et al. (2017) analyzed a series of chemistry-climate model experiments conducted for the period January 1960 through 2014. They found structural

changes in BDC have occurred in the BDC since 1980s; BDC accelerated in the lower stratosphere in the northern hemisphere and tropics but not in the mid-stratosphere, and specifically since 1992, mean age increased by 0.12 year in the mid-stratosphere of the northern hemispheric mid-latitude and tropical mass upwelling has slowed down by 2 %. As discussed in our previous study (Sugawara et al., 2018), gravitational separation of the stratospheric air is sensitive to changes in mean meridional circulation rather than horizontal mixing. Therefore, it is thought that the increase of mean age in the mid-stratosphere in the

northern mid-latitude and the slowdown of tropical upwelling, suggested by the recent observational and modeling studies, are consistent with our weakened BDC simulation of the $\delta(Ar/N_2)_\Omega$.

  We also carried out an additional simulation of $\delta(Ar/N_2)_\Omega$ considering the interannual variation in the BDC, to examine its effect on the large interannual variations of the observed $\delta(Ar/N_2)$ seen in Fig. 4. As the BDC interannual variation, we simply assume a 10 % change in the MMC intensity with a 3-year cycle by changing the mass stream function in the 2-D model. Flury

et al. (2013) reported that the speeds of BDC towards the NH and SH show interannual variabilities with amplitudes of about 21 and 10 %, respectively, and that the amplitude of variability in the ascent rate at the Equator is 21 %. Therefore, the 10% change in the MMC in our simulation is not an unrealistic assumption. As a result, the increase rate of $\delta(Ar/N_2)_\Omega$ showed an interannual variation with a peak-to-peak amplitude of $\pm0.4$ per meg $yr^{-1}$. This interannual variation is non-negligible but still rather small compared to that found in the observed $\delta(Ar/N_2)$, about $\pm4.5$ per meg $yr^{-1}$, as seen from Fig. 4. Therefore, it is

suggested that the interannual variation of surface $\delta(Ar/N_2)$ is driven mainly by solubility change in seawater and/or that the assumed $\pm10$ % changes in the MMC intensity in our model is smaller than the actual interannual variation of the MMC. In addition, analytical artifacts and the extremely difficult challenge of maintaining stable standard air for $\delta(Ar/N_2)$ could also be cause(s) of the interannual variation. Therefore, further studies are needed to achieve quantitative understanding of the discrepancy.

Consequently, if we regard $OHC_{oc}$ as representing "true" global OHC, then we can estimate secular trend and interannual variations in the BDC from $\delta(Ar/N_2)$ observed at the surface. Conversely, if the simulated $\delta(Ar/N_2)_\Omega$ represents "true" BDC changes, then we can validate $OHC_{oc}$ by using $OHC_{ArN2}$. Therefore, the surface $\delta(Ar/N_2)$ is found to be a unique tracer for detecting changes in a spatiotemporally-integrated OHC and BDC. Not only precise observations of $\delta(Ar/N_2)$ over longer periods of time, but also improvements in $\delta(Ar/N_2)_\Omega$ simulation using 3-D atmospheric transport models are important for

future research advances in this field. For example, recently, Birner et al. (2020) simulated lower stratospheric $\delta(Ar/N_2)$ using the TOMCAT/SLIMCAT 3-D chemical transport model, which has been updated to include gravitational fractionation of gases. Such activities will contribute significantly to understanding the mechanisms of spatiotemporal variations in the stratospheric and the surface $\delta(Ar/N_2)$.

**4 Conclusions**

We have been carrying out systematic measurements of atmospheric $\delta(Ar/N_2)$ at TKB and HAT since 2012, COI and SYO since 2013 and 2016, respectively. Clear seasonal cycles of $\delta(Ar/N_2)$ with summertime maximum were found at TKB, COI and SYO, with peak-to-peak amplitudes of $21\pm10$, $11\pm4$, $5\pm10$ and $32\pm9$ per meg at COI, TKB, HAT and SYO, respectively, which are in general agreement with those observed at similar latitudinal sites by other investigators. The observed $\delta(Ar/N_2)$ at TKB and HAT, after subtracting seasonal cycles and shorter-term variations, showed significant interannual variations and

slight, but detectable, secularly increasing trends. The observed interannual variations correlated positively with that of the

global OHC, suggesting a strong correlation between large-area air-sea heat flux and the long-term change in $\delta(Ar/N_2)$. However, the ratio of interannual variation to secular increase is found to be much larger in $\delta(Ar/N_2)$ than that in OHC, so further studies are needed to interpret the discrepancy.

We improved the 2-D model we used in previous studies to calculate gravitational separation in order to evaluate its effect on
the long-term change in the surface $\delta(Ar/N_2)$. We simulated weakened and enhanced BDC by arbitrarily changing the MMC (represented by mass stream function in the model), and obtained effects of 0.15 and -0.13 per meg yr$^{-1}$, respectively, on the secular trend of the surface $\delta(Ar/N_2)$. If we apply the correction for gravitational separation to the secular trend of $\delta(Ar/N_2)$ observed at TKB, then an average increase rate of 0.60±0.30 per meg yr$^{-1}$ for $\delta(Ar/N_2)$ during 2012-2019 was obtained by assuming a weakening BDC. By using a conversion factor of 3.5 (3.0) x10$^{-23}$ per meg J$^{-1}$ by crudely assuming a 1-box ocean
with a temperature of 3.5 (7.5) °C, we obtained an increase rate in the global OHC of 17.1±8.6 (20.2±10) ZJ yr$^{-1}$ for the 8-year period, which is consistent with 12.2±1.2 ZJ yr$^{-1}$ reported by NOAA/NCEI from the Argo float observations. On the other hand, the increase rate in the global OHC, obtained from the secular trend of $\delta(Ar/N_2)$ at TKB under enhanced BDC, was found to be significantly larger than that from the Argo float observations. These results indicate that the surface $\delta(Ar/N_2)$ is a unique tracer for spatiotemporally-integrated OHC and BDC. Although an increase in global OHC is well known as an essential
parameter to evaluate recent global warming, there is no method yet to adequately measure OHC via ocean temperature observations in the full-depth volume of the ocean. Long-term precise observations of $\delta(Ar/N_2)$ will meet this demand, after correction for gravitational separation of the whole atmosphere.

## 5 Appendix A: Additional description of the SOCRATES model

We performed numerical simulations using a 2-dimensional model of the middle atmosphere (SOCRATES) developed by the National Center for Atmospheric Research (NCAR) (Huang et al., 1998; Park et al., 1999; Khosravi et al., 2002). Since details of our model calculation for gravitational separation have already been described in Ishidoya et al. (2013) and Sugawara et al. (2018), only a brief description is given here. The model extends from the surface to 120 km altitude with a 1 km vertical resolution, and from 85°S to 85°N latitude with a 5° latitudinal resolution. Time step of this model can be varied to
accommodate chemical transport equations of various species. However, in our study, we used a time step of 1 day, because Ar and $N_2$ molecules have no chemical reaction and the time constant of the molecular diffusion process is sufficiently longer than 1 day. The meridional and vertical eddy diffusivity coefficients ($K_{yy}$ and $K_{zz}$) are parameterized by including three types of waves: planetary, gravity and tidal waves. The mass transport processes caused by molecular diffusion were originally taken into account only above the mesosphere in SOCRATES, since the molecular diffusion effect was thought to be negligibly
small in the troposphere and stratosphere, compared with the eddy diffusion effect. But since our study needed to include the effect of molecular diffusion in the lower atmosphere, however small compared to eddy diffusion, we simply lowered its

vertical domain to the surface for the calculation of molecular diffusion. Molecular diffusion theory included in SOCRATES is based on Banks and Kockarts (1973).

## 6 Appendix B: Performance of the updated SOCRATES model

The annual mean meridional distribution of the $\delta(Ar/N_2)_\Omega$ value calculated using the updated SOCRATES model is shown in Fig. A1. The model shows that the vertical $\delta(Ar/N_2)_\Omega$ gradient increases with increasing height, especially at high latitude, which is consistent with the meridional distributions reported by Ishidoya et al. (2013) and Sugawara et al. (2018) calculated using the original SOCRATES model. The basic structure of the vertical–meridional $\delta(Ar/N_2)_\Omega$ distribution can be interpreted as a result of a balance between the mass-dependent molecular diffusions and the mass-independent transport processes. Figure A2 shows the vertical $\delta(Ar/N_2)$ profiles observed over Sanriku (39°N, 142°E), Japan on June 4, 2007 (Ishidoya et al., 2013) and Biak (1°S, 136°E), Indonesia on February 22–28, 2015 (Sugawara et al., 2018). The vertical $\delta(Ar/N_2)_\Omega$ profiles calculated by the updated SOCRATES model for the same seasons are also shown. As seen from the figure, the average vertical gradients of the observed vertical $\delta(Ar/N_2)$ profiles are generally reproduced by the calculated $\delta(Ar/N_2)_\Omega$, although the fluctuations in the observed profiles are not simulated. These fluctuations are also observed over Antarctica and could be attributed to the horizontal mixing of the stratospheric air over the region (Ishidoya et al., 2018), their occurrences at low and middle latitudes are still not well understood. Overall however, we are satisfied with the performance of the updated SOCRATES model in reproducing the average vertical gradient of the $\delta(Ar/N_2)$ profiles from the surface to the stratosphere.

Figure A3 shows the annual mean meridional distribution of the AoA secular trend calculated using the updated SOCRATES model for weakened and enhanced BDC simulations. The simulations were made by arbitrarily changing the mass stream function in the model so that AoA changed by +0.02 (weakened BDC) or -0.02 (enhanced BDC) yrs yr$^{-1}$ at 35 km over the northern mid-latitudes. The AoA trends are larger in the middle stratosphere than those in the lower stratosphere for the weakened BDC simulation. This altitudinal increase of the AoA trend is consistent with that derived from the ERA-Interim (Diallo et al., 2012), however, the updated SOCRATES model cannot reproduce the negative AoA trend in the lower stratosphere found in Fig. 13 of Diallo et al. (2012). Ray et al. (2014) also reported increase and decrease of AoA in the middle and lower stratosphere, respectively. Therefore, simulations of the BDC secular trend using the updated SOCRATES model do not fully represent the details of the meridional distribution in the real atmosphere. Further studies are needed by using 3-D models for validating the $\delta(Ar/N_2)_\Omega$ secular trend discussed in the present study.

*Data availability.*

The observational data of $\delta(Ar/N_2)$ presented in this study can be accessed by contacting the corresponding author.

*Author contributions.*

SI designed the study, carried out $\delta(Ar/N_2)$ measurements and drafted the manuscript. SS improved 2-D model and carried out $\delta(Ar/N_2)_\Omega$ simulations. YT and DG managed the collections of air samples at Hateruma, Ochiishi, and Syowa station. KI examined the relationship of Ar and $N_2$ fluxes with air-sea heat flux. NA prepared gravimetric standard mixtures of Ar, $O_2$, $N_2$ and $CO_2$. SS, YT, DG, KI, YN, NA and SM examined the results and provided feedback on the manuscript. All the authors approved the final manuscript.


*Competing interests.*

The authors declare that they have no conflict of interest.

**Acknowledgements.**

We thank staff of Global Environmental Forum (GEF) and the Science Program of the Japan Antarctic Research Expedition (JARE) for their works to collect the air samples at Hateruma and Ochiishi stations (GEF) and Syowa station (JARE). This study was partly supported by the JSPS KAKENHI Grant Number 15H02814 and 18K01129, and the Global Environment Research Coordination System from the Ministry of the Environment.

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

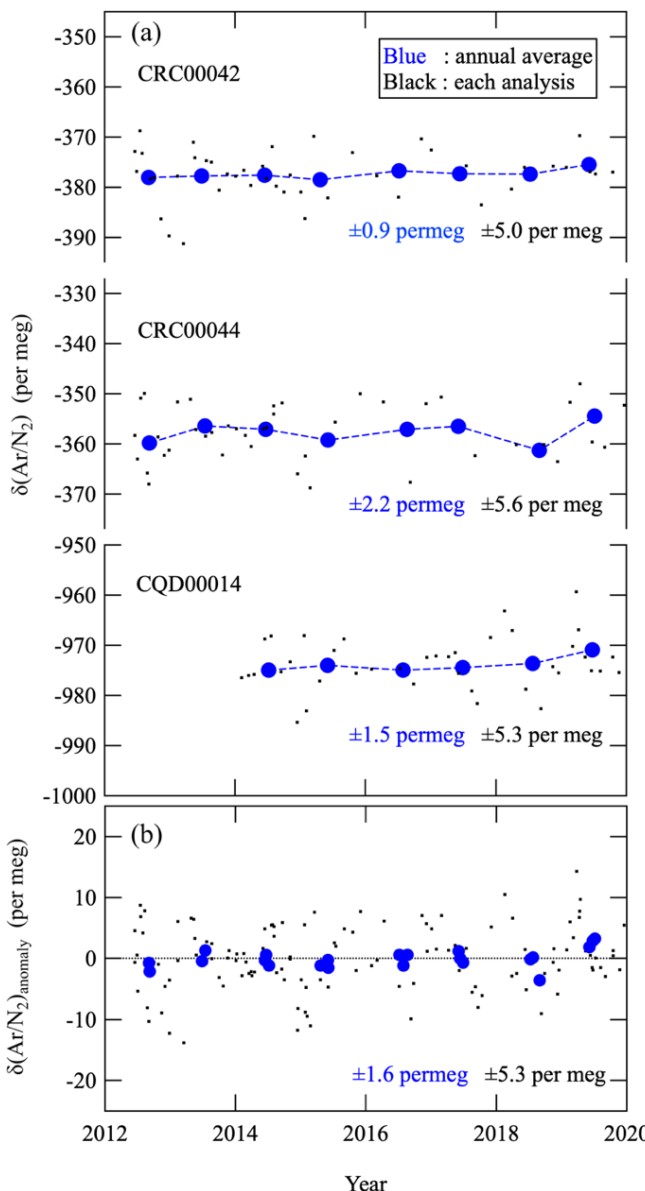

**Figure 1: (a) Each analysis value (black dots) and corresponding annual average (blue circles) of $\delta(Ar/N_2)$ of 3 standard air against the primary standard air. (b) Anomalies of $\delta(Ar/N_2)$ of 3 standard air shown in (a).**

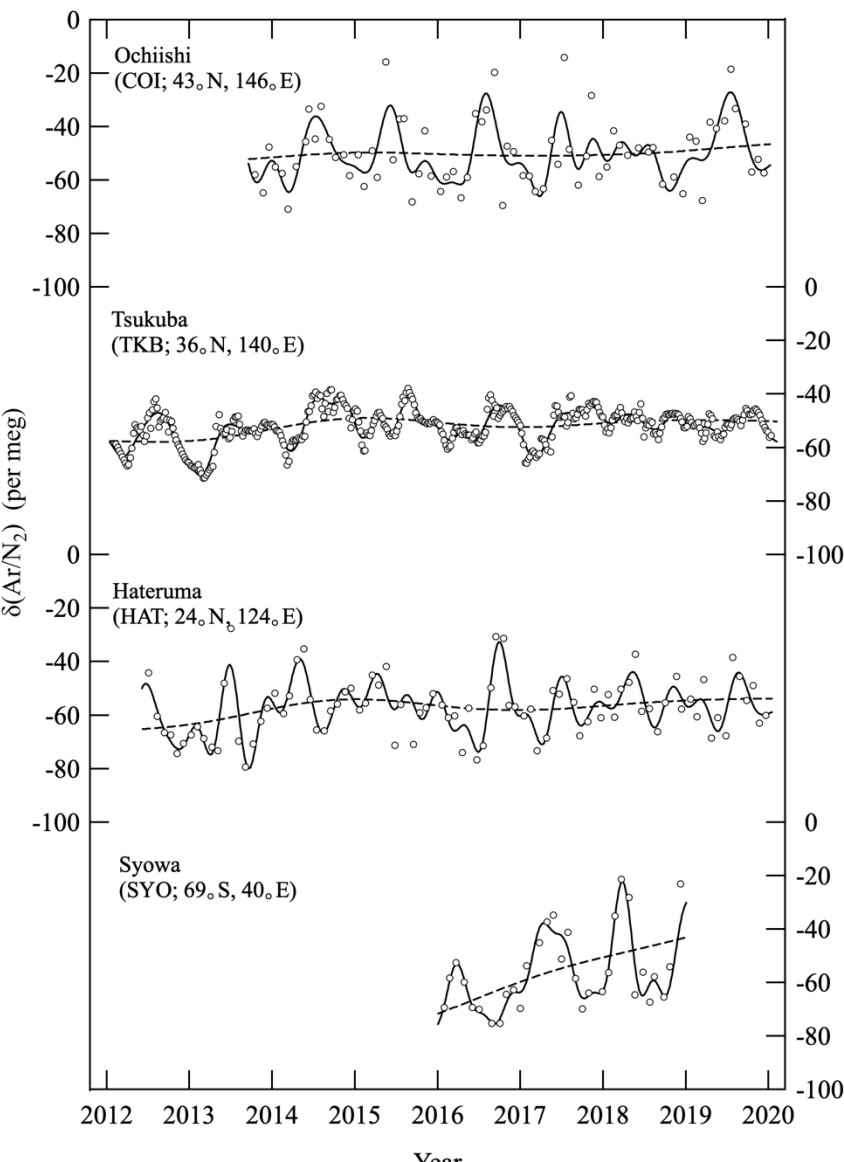

**Figure 2: Temporal variations of δ(Ar/N₂) at Ochiishi (COI), Tsukuba (TKB), and Hateruma (HAT), Japan and Syowa (SYO), Antarctica. Observed values are shown by open circles. Best-fit curves to the data (solid lines) and interannual variations (dashed lines) are also shown.**



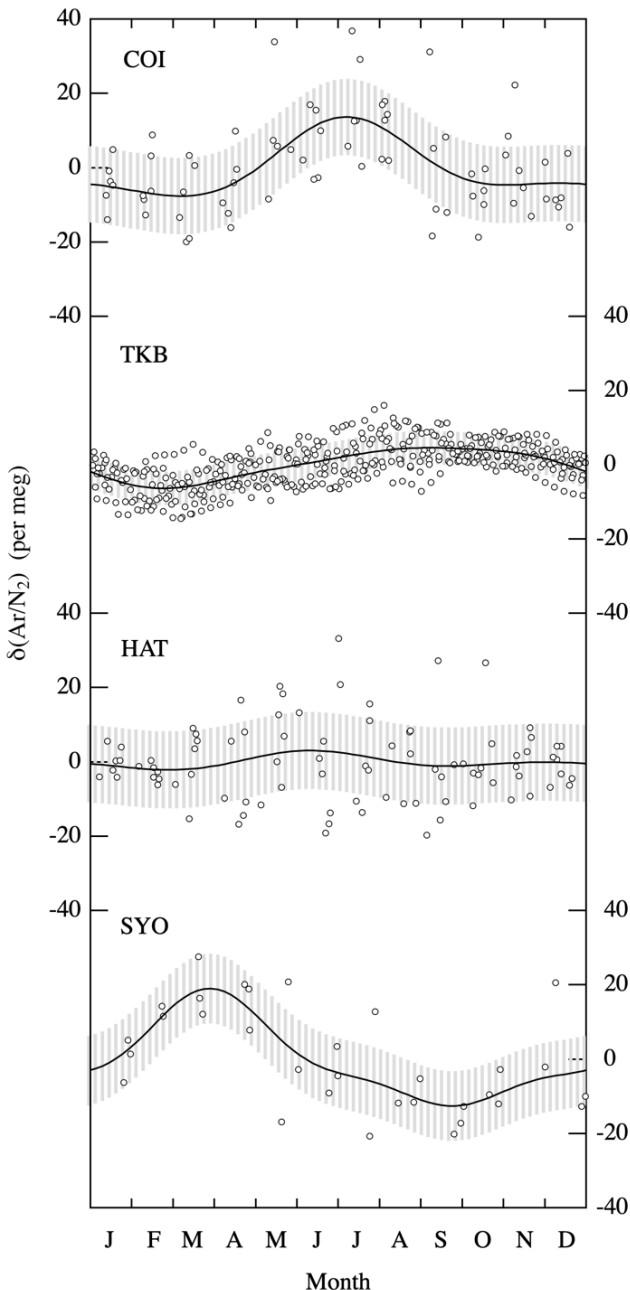

**Figure 3: Detrended values of δ(Ar/N₂) (open circles) and average seasonal cycles of δ(Ar/N₂) (solid lines) observed at COI, TKB, HAT and SYO. Gray shaded areas denote standard deviations of the detrended values from the average seasonal cycle at each site.**

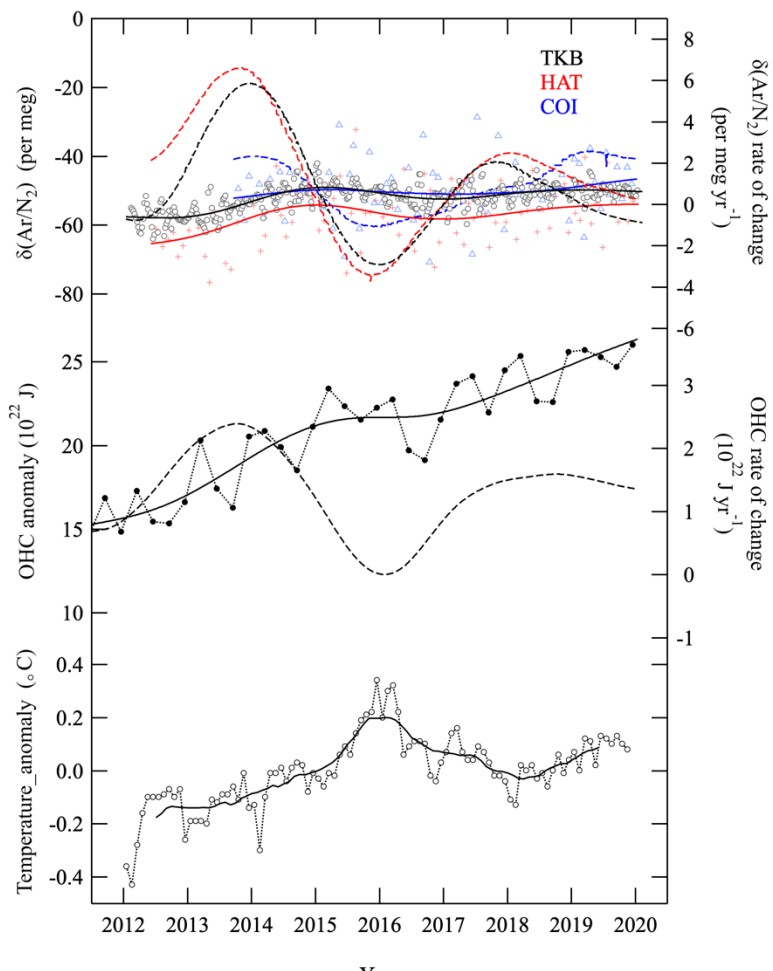

Figure 4: (top) δ(Ar/N₂) values at TKB (black open circles), HAT (red crosses) and COI (blue triangles), after subtracting seasonal cycles and shorter-term variations less than 36 months. Interannual variations of δ(Ar/N₂) at TKB, HAT and COI are shown by black, red and blues solid lines, respectively. Rates of change of δ(Ar/N₂) at TKB, HAT and COI are also shown by black, red and blue dashed lines, respectively. (middle) 0-2000 m global ocean heat content (OHC) from NOAA/NCEI (filled circles), its long-term trend (solid line) and increase rate (dashed line). The baseline period for OHC is mid-1980s. (bottom) Global average surface temperature anomalies. Monthly and their 12-months running mean values are shown by open circles and solid line, respectively.



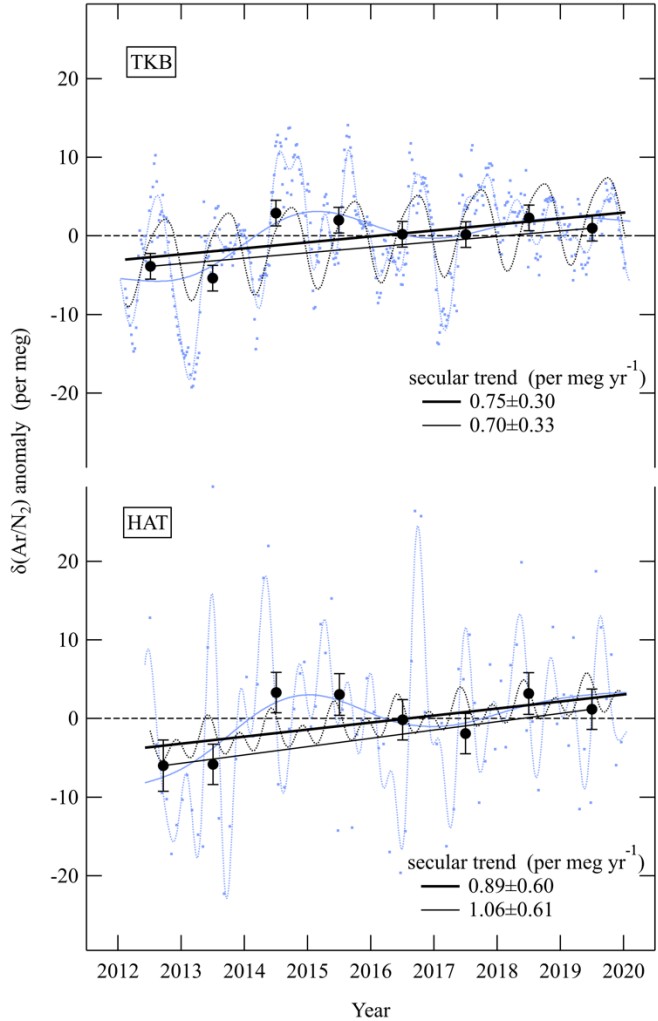


**Figure 5: Same observed values (blue dots), best-fit curves to the data (blue dotted lines) and interannual variations (blue solid lines) of δ(Ar/N₂) at TKB (top) and HAT (bottom) as in Fig. 2. Data are expressed as anomalies from the average value for the observation period. Another best-fit curves consisting of the fundamental, its first harmonics and a linear secular trend to the observed data are also shown (black dotted lines). The components of the linear secular trend of the best-fit curve are shown by thick black solid lines.**

**In order to address the uncertainty of the secular trend, uncertainties around the regression and the long-term stability of the standard air (±1.6 per meg, blue circles in Fig. 1 (b)) have been taken into account. Annual average values of δ(Ar/N₂) at TKB and HAT (filled black circles), and secular trend at each station calculated as the difference between the 2012 and 2019 annual average values are also plotted (thin black solid lines). Error bars for the annual average values at TKB and HAT are evaluated considering**

the uncertainty of each analysis value of the standard air (black dots in Fig. 1 (b))  and the uncertainty associated with repeated

analyses of the same air samples.


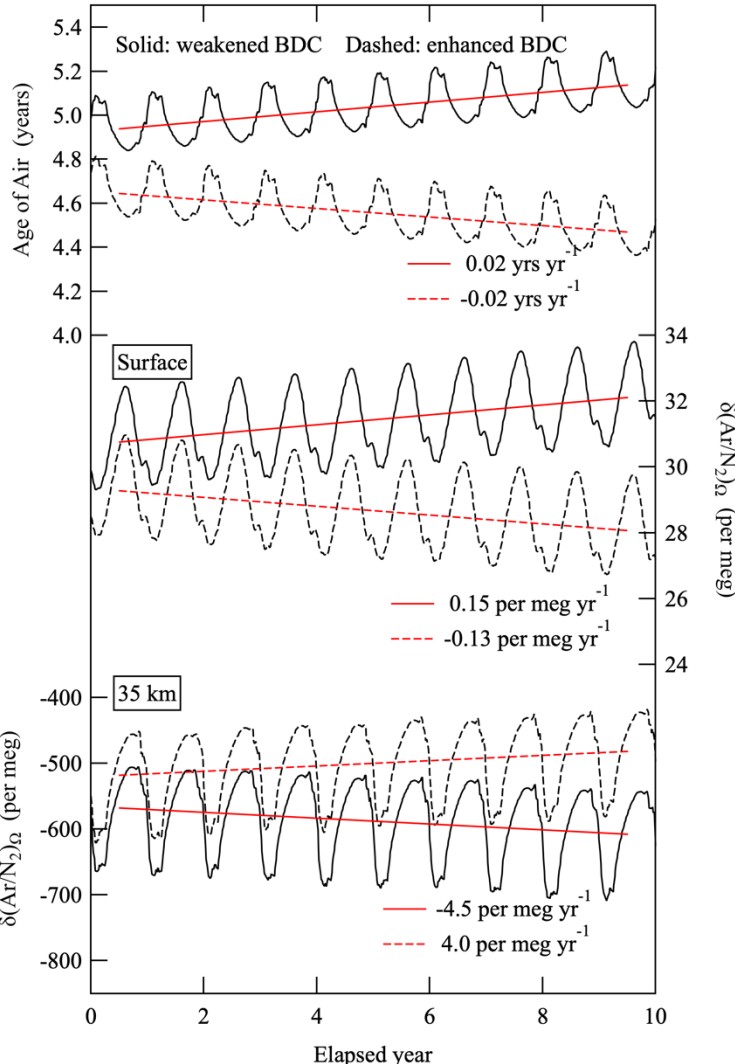

**Figure 6: Age of air (AoA) in the stratosphere at 35 km, $\delta(Ar/N_2)_\Omega$ at the surface and $\delta(Ar/N_2)_\Omega$ at 35 km simulated by using the updated SOCRATES 2-D model for weakened and enhanced BDC conditions (see text). Red solid and dashed lines denote secular trends of $\delta(Ar/N_2)_\Omega$ for the weakened and enhanced BDC simulations, respectively, obtained by applying linear regression analyses to the annual average data.**


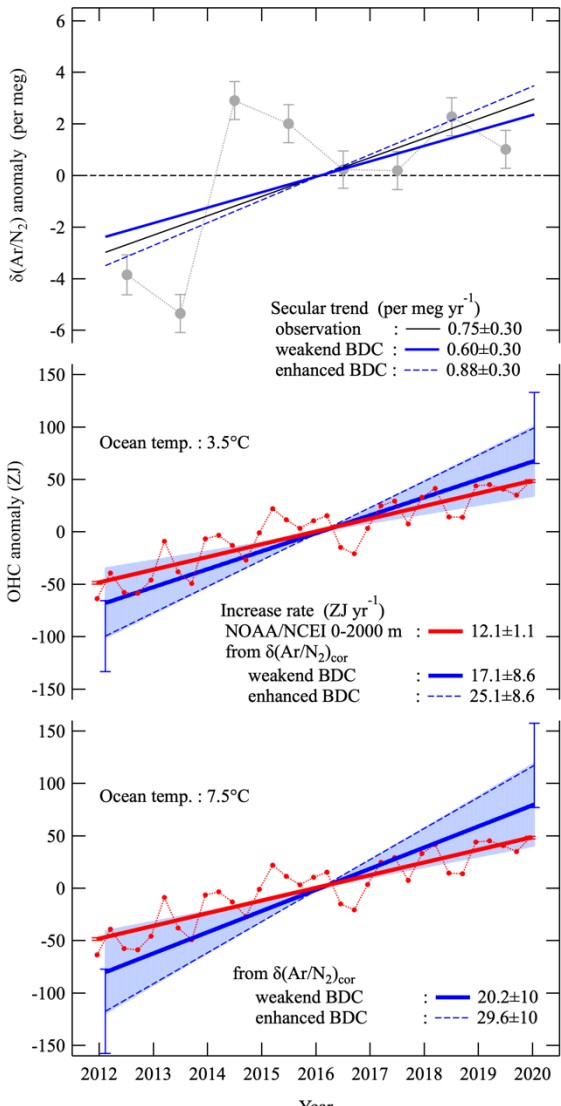


**Figure 7: (top)** Same annual average values (gray filled circles) and secular trend (black solid line) of δ(Ar/N₂) observed at TKB (from Fig. 5). Thick blue and dashed blue lines denote the secular trends of δ(Ar/N₂)cor derived by subtracting the secular trends of δ(Ar/N₂)Ω expected from weakened and enhanced BDC simulations, respectively, from the linear trend of δ(Ar/N₂) at TKB. **(middle)** Global OHC increase estimated from the secular trends of δ(Ar/N₂)cor for weakened and enhanced BDC conditions (thick blue and dashed blue lines, respectively) using a conversion factor of $3.5 \times 10^{-23}$ per meg J$^{-1}$ by assuming a 1-box ocean with a temperature of 3.5 °C. Light blue shade and blue error bars denote the uncertainties of the OHC increases estimated from δ(Ar/N₂)cor for weakened and enhanced BDC conditions, respectively. Red circles and the regression line denote the secular increase of 0-2000 m global OHC from NOAA/NCEI. **(bottom)** Same as in middle but for using a conversion factor of $3.0 \times 10^{-23}$ per meg J$^{-1}$ by assuming a 1-box ocean with a temperature of 7.5 °C.

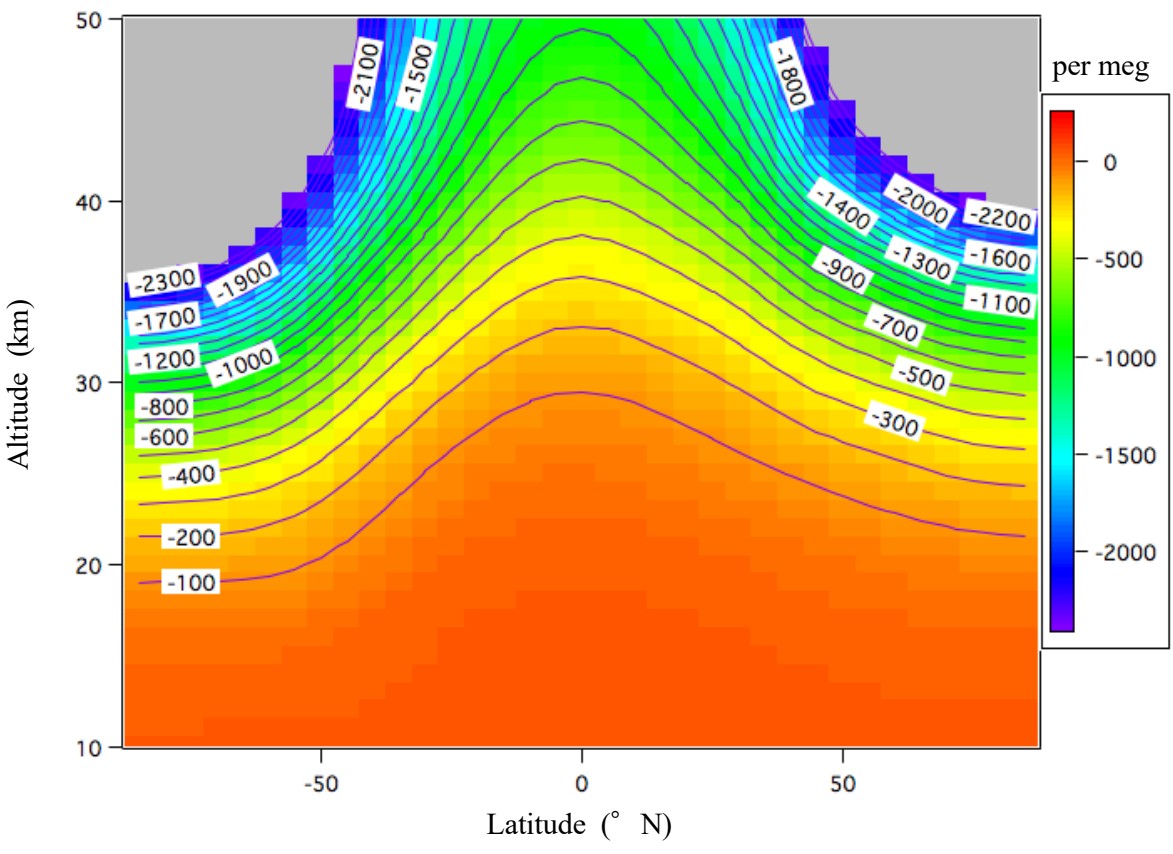


**Figure A1: Annual mean meridional distribution of δ(Ar/N₂)$_Ω$ calculated using the updated SOCRATES model for the steady state condition. The values lower than −2500 per meg are shown in gray.**

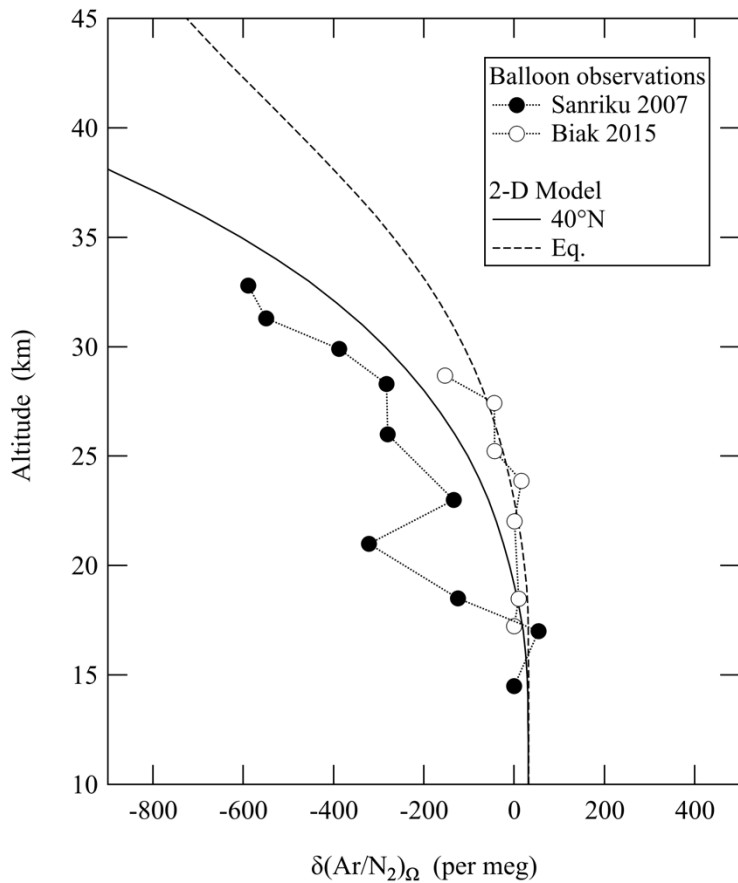


**Figure A2: Vertical profiles of δ(Ar/N₂) observed over Sanriku (39°N, 142°E), Japan on June 4, 2007 (Ishidoya et al., 2013) and Biak (1°S, 136°E), Indonesia on February 22–28, 2015 (Sugawara et al., 2018). Vertical profiles of δ(Ar/N₂)$_\Omega$ calculated by the updated SOCRATES model for the same seasons are also shown. Observed values are expressed as deviations from the measured values at 14.5 km over Sanriku.**


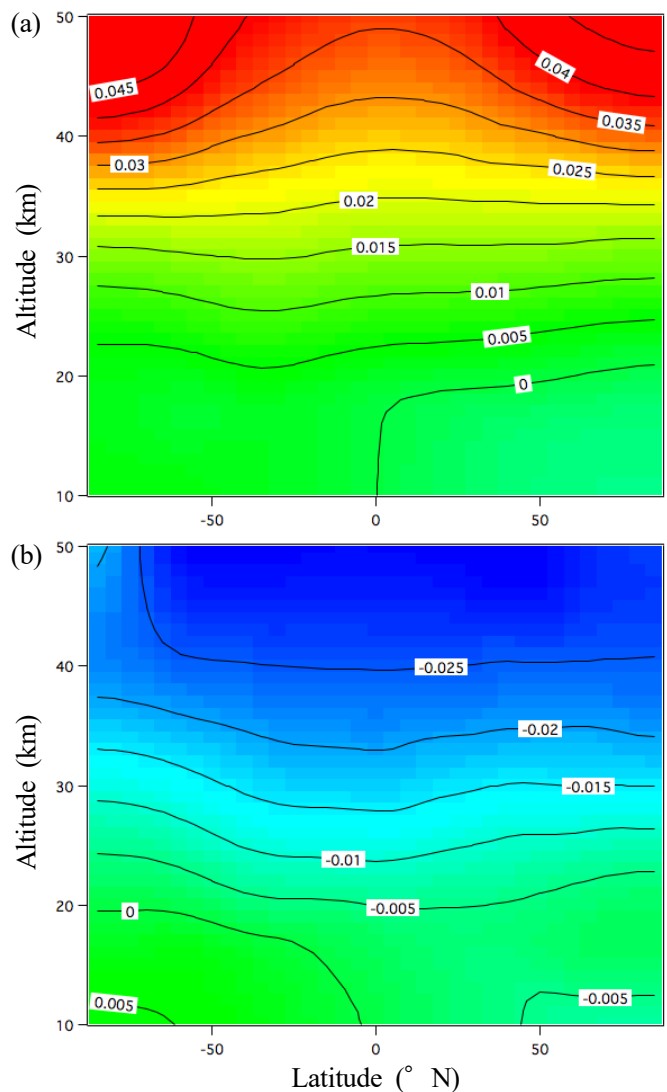


**Figure A3: (a) Annual mean meridional distribution of AoA trend (yrs yr⁻¹) calculated using the updated SOCRATES model for weakened BDC simulation. (b) Same as in (a) but for enhanced BDC simulation.**