# Peer review of "Secular change in atmospheric $Ar/N_2$ and its implications for ocean heat uptake and Brewer-Dobson circulation"

_Atmospheric Chemistry and Physics, 2020_

## Referee Comment (RC1) · Anonymous Referee #1 · 5 Jul 2020

This paper presents observations of surface Ar/N2 ratios from four sites (3 in Japan and 1 in Antarctica). A long-term trend is diagnosed from the two sites with the longest records (2012-2020) and this is converted to a trend in global ocean heat content. A 2-D atmospheric model is used to estimate the trend in surface Ar/N2 that might be caused by an assumed trend in the stratospheric Brewer-Dobson circulation, which amounts to 20% of the overall Ar/N2 trend. The corrected surface trend (and the uncorrected one) are consistent (within the large error bars) with the OHC trend calculated from temperature data.

I think that the paper presents interesting observations and is attempting a novel anal-

ysis. Investigating how surface Ar/N2 can be used to quatify ocean heat uptake is an important topic and investigating how stratospheric variations can influence this is timely. However, I find that some aspects of the methods are crude and/or not fully explained. I also think that caveats in the analysis need to be stated. Therefore, I think that the paper requires major changes before becoming possibly suitable for publication in ACP. My comments are below.

Main Points

1) Surface observations. TKB has the longest and most dense data record. The observed temporal variations (Figure 2) give a compact annual cycle. In contrast, the sparser observations at the other 3 sites show much more variability (large amplitude variations). What are the reasons for this? Is it a measurement issue or possible real atmospheric variations?

2) Observed long-term trend. Even the two longest data records (8 years) are short for deriving accurate long-term trends. The trend fit is not explained clearly. The model leaves variations which are > 36 months but the trend is quoted as if it is a linear term? The fits to Ar/N2 and OHC in figure 2 are far from linear (but do vary with the long-term variability in the temperature data). I am concerned that the paper is reporting values in the abstract which imply a long-term linear trend, which is (quantitatively) not obvious from the plots.

3) Atmospheric modelling. An updated 2-D model has been used to model surface Ar/N2. There is no evaluation of the model in the stratosphere using available profiles of Ar/N2 to show that the higher altitude gravitational separation is modelled realistically.

4) Imposed trend in the BDC. The stratospheric BDC is complicated with deep and shallow branches. A trend in the circulation is imposed in the model and the resulting trend at 35 km is shown. First, more information on the impact at other altitudes should be shown (e.g. latitude-height cross section of the impact on age-of-air). Second and more importantly, the use of the Engel et al paper to support a trend in AoA, which is

converted to a correction of surface Ar/N2 is unjustified. Engel et al use a series of sparse balloon observations of CO2 and SF6 to derive an AoA trend (from 24-35km) up to 2005 – so there is no overlap with the observation period in this paper. Moreover, the error bar on their trend is very large and the title of their paper gives the headline message of 'AoA unchanged within uncertainties'. Therefore, I cannot see how the imposed trend of 0.02 yrs/yr can be justified as the best estimate which gives the correction used in the abstract.

5) Ocean model. This is a crude approach (as acknowledged by the authors) and it leads to statements in lines 187-193 that the model maybe too simplistic (i.e. not suitable) and that other factors may need to be considered. Overall, this part of the analysis seems incomplete therefore.

Other Specific Points

Line 21. The uncorrected trend of 0.75 +/- 0.30 per meg yr-1 is also consistent with trend derived from ocean temperature, at the limits of the error bars. The correction is not needed, which is what is implied by the text.

Line 132. 'Fig 2 in 3-1'. What does this mean?

Line 163. 'Increase rates' (also in caption of Figure 4 and label axis). This should be referred to simply as 'rate of change'. The positive values will imply an increase.

Line 180 'boldly'. This is the wrong word. You must mean something like 'crudely'.

Line 183. 'modern' – better to say 'present-day'.

Line 184. Insert 'total atmospheric mass….'.

Line 187 'drives'. Delete s

Line 216. 'molecular mass number' -> 'relative molecular mass'

Line 221-223. Add a reference or make it clear that you are referring to this work.

Line 238. 'decrease' (no s)

Line 240 (and later). I think 'idealised' is better than 'virtual'.

Line 250. Don't need to say 'increase'.

Line 251. Give the dates that the Engel et al study covered (but see main comment above).

Line 255. The word 'obtained' is wrong. The perturbation to the model circulation was forced arbitrarily. Change to e.g. 'forced'?

Line 258. Nb 'simplistic' is a negative term which means that the model is too simple to be suitable.

Lines 265. Change 'increases' to 'is estimated to increase'.

Line 267. Change 'expected' to 'is estimated'.

Line 269. Insert 'are non-negligible trends compared. . .'

Line 274 Insert '. . . estimated effects..'

---

## Referee Comment (RC2) · Anonymous Referee #2 · 7 Jul 2020

Ishidoya et al. present an analysis of atmospheric variations of the Ar:N2 ratio. Changes in oceanic heat content (OHC) drive variations in atmospheric Ar:N2 due to changes in solubility. Measurements of Ar:N2 (reported in delta-notation relative to a standard) thus bear promise to infer OHC changes without measuring oceanic temperatures. Given the paramount role of OHC in the context of global warming, complimentarty methods that provide independent estimates of OHC changes such as the one discussed in this paper are extremely important.

The study by Ishidoya et al. argues that the Ar:N2 ratio measured near the surface not only reflects OHC changes, but also changes in the strength of the stratospheric

[Figure]

Brewer-Dobson circulation. The latter is a consequence of non-negligible gravitational separation according to earlier work by the same group. In a nutshell: The atmosphere is depleted of Ar with increasing height. A strengthening of the BD circulation decreases the separation, leading to an Ar:N2 ratio increase aloft, and decrease near the surface.

The manuscript is challenging to review as it touches on a wide range of topics but information provided is often (too) sparse. As far as I can see, the main points of the manuscript are:
(i) Ar:N2 measurements are accurate enough to detect trends, which was not possible previously (Line 37).
(ii) Variations in the ratio over a decade are well correlated with OHC variations as reported by NOAA/NCEI.
(iii) There is a quantitative mis-match if only OHC is considered.
(iv) A model simulation with imposed change in the stratospheric BD circulation suggests that the near surface Ar:N2 ratio is as sensitivite to the BD circulation as to OHC variations (for the observed magnitude of changes in either of these).
(v) If one assumes that the OHC measurements and their impact on Ar:N2 are correct, one infers that over the period of measurements (2012-2019) the BD circulation slowed down.

The authors discuss the last point in some detail, and seem to be concerned that the sign of the change in the BD circulation is the opposite of a supposedly long term strengthening of the BD circulation. There is no reason, however, why a short period such as the one studied here should show the long term trend given the large interannual variability of the stratospheric circulation. I recommend to shorten this somewhat misleading discussion (the authors are most likely not observing "trends"), and instead strengthen the points (i)-(iv). Specifically:
(a) This reviewer is not sufficiently familiar with the measurment technique to be able to assess claim (i); the data as presented in Figures 2,3 and 4 does look, however, reasonable; as does the calibration (Figure 1). Nonetheless, I hope that another reviewer may be in a position to comment.

(b) The time filtering and frequency separation method needs to be better explained - based on the information provided in this manuscript, it is not possible to reconstruct what precisely the authors have done. This is an extremely important part of the argument as the claim is that seasonal variations are due to local SST variations, whereas the longer-term changes are due to global OHC changes (i.e. the seasonal cycle at TKB and HAT differ (Fig 3), but their longer-term variations (Fig 4) are highly correlated). Also, the analysis then uses largely only TKB and HAT - but why not also show COI in Figure 4? Please use 2 different colors for AHT and TKB in Figure 4a - the separation with "+" and "o" does not allow to see the differences (Looking at Figure 2; it is surprising that these two timeseries align as well as shown in Figure 4).

(c) The SOCRATES model is not sufficiently described. Please provide more information on vertical and horizontal resolution, time steps, and parameterization of atmospheric transport and mixing. Please show the model's mean vertical profile of the Ar:N2 ratio, along with the measurements thereof in the stratosphere. Do you have other information (e.g. CO2 mixing ratio profile) that could serve as validation?

(d) Generally, a better description of the system studied, and implicit assumptions, would be helpful. (Also - equations 3 and 4 can be valid only for specific conditions - please provide more details).

Additional points:

- Figure 5: My understanding is that this is simply an idealized calculation for 10 years; if so, please change x-label (labels 2000 - 2010 suggest that this is specifically for this period; which is confusing also because OHC changes are taken from this period, see next item).

- OHC: Why do you discuss OHC change for 2000-2010 (Line 267)? Why are you not using the data for 2012-2019? The text here is confusing; Figure 4b suggests you use the data for the correct period.

- Please indicate the baseline period for OHC (i.e. the figures shows departures from

a baseline, not absolute OHC as one would think based on labels and caption).

---

## Referee Comment (RC3) · Anonymous Referee #3 · 27 Jul 2020

This is an excellent paper. It breaks open a whole new idea: namely that the stratosphere's well-known gravitational separation (in which heavy gases such as argon settle out relative to lighter gases such as N2), may have a small but perceptible impact on the Ar/N2 ratio of the troposphere, by the simple logic of whole-atmosphere mass conservation. Variations in the amount of gravitational settling over time, due for example to variability in the Brewer-Dobson circulation in the stratosphere, are proposed to have a very small impact of opposite sign in the troposphere. Excellent quality long-term data from air monitoring stations is shown to back up this claim. However, it remains to be demonstrated that the signals seen in the data are entirely due to the stratospheric effect; other minor processes such as imperfect mixing within the troposphere

and shallow-ocean temperature anomalies may play a partial role, and so further work is needed to track down these possible issues. But the authors have made a very important contribution and this paper should be published with only minor revisions.

The authors furthermore expand and improve on the Blaine and Keeling approach to decadal-scale ocean heat content variations deduced from tropospheric Ar/N2 measurements. To my eye, this is the best Ar/N2 data ever published on this critical topic of ocean heat, and the authors are to be commended for this important contribution to understanding the Earth's net heat budget under the current anthropogenic forcing. These type of data may eventually make it possible to better constrain the climate sensitivity to a doubling of atmospheric CO2, because the ocean may be masking the true top-of-the atmosphere energy imbalance, due to the fact that something like 93% of this excess energy goes into the ocean and so may be escaping our detection. The present paper is an important step in this fundamentally critical direction.

The one minor comment I have is that the one-box ocean model is somewhat over-emphasized, and perhaps the description could be simplified and shortened, because we know very well that the troposphere does not perfectly mix on a timescale of a year, nor does the ocean mix perfectly on a timescale of one year, so it is not really surprising that a one-box ocean model fails to match the observations of Ar/N2 at surface stations around Japan. I understand that the authors constructed the one-box model as a "straw man", to be shot down, but they could greatly simplify and shorten the discussion, while making it clear that they do not expect to be able to match their observations with a one-box model of the ocean. In fact, some readers may be confused, the way the authors have written about this one-box model (they seem to imply that they expected it to be able to match their observations). But of course the surface-ocean temperature anomalies are by far the largest source of noise on yearly timescales, for tracers like Ar/N2, measured in near-surface air. They should clarify that a true tropospheric-average Ar/N2, measured from aircraft (which is of course too expensive and so is prohibitive), would be needed to actually compare Ar/N2 observations with modeled

Ar/N2. (So in some sense they are raising a false expectation with their one-box model exercise.)

On the whole, this is a groundbreaking and important paper, and will make a fine contribution to ACP. This is clearly excellent, highest-quality work!

Minor editorial comments:

line 35 ". . ..Argo floats"

line 47 - ". . .since it has been believed that the gravitational separation near the surface is too small to be detected. . ." This isn't really accurate, in the sense that there is no need for the gravitational settling process IN THE troposphere to be significant, in order for the stratosphere to affect the troposphere. Perhaps instead you could say something like, ". . .since it has been believed that the gravitational settling in the stratosphere is fairly small and constant in time, along with the fact that the troposphere has 10x more molecules than the stratosphere."

Or perhaps you meant to say, ". . .the gravitational separation signal from the stratosphere is too small to be detected at the surface."? This is accurate.

line 51 ". . .long-term changes in the Ar/N2 ratio near the surface are expected to be extremely small.."

line 53 again, it sounds like you are saying there there is gravitational separation near the surface, but this is not accurate. Maybe you mean the stratospheric gravitational separation signal near the surface?

line 58 ". . .secular trend of the Ar/N2 ratio. . .."

line 60 "Atmospheric Ar/N2 has been observed. . . .." (there is no need to include "ratio" here)

line 74 ". . ..and we usually use the average of 550 data values as the reported Ar/N2 ratio obtained from the continuous observations (about 11 hours of averaged data)."

line 101 "…glass flasks."

line 102 "per meg units as follows."

line 112 "… corresponds to an uncertainty of…."

line 116 "…to +1,800"

line 122 "..but they did not correlate.."

line 124 "…must have been superimposed…"

line 126 "…due to a temperature…."

line 138 "…by a fundamental sine-cosine…"

line 143 "..reach seasonal maxima …..  due to the larger relative…" (enhancing is not really the correct word, since the solubility is a physical constant and cannot be enhanced)

line 146 "Similar increases…."

line 150 "..found a seasonal…"

line 154 "…than the 14±6…."

line 160 this sentence is very long and hard to read. Perhaps you could simplify and shorten it. Also the verb "are shown" comes at the very end of the sentence, which is awkward. Instead you could write, "Variations in the 0-2000 m global OHC are shown (Fig. 4), reported by…."

line 180 "We boldly modeled…." This is not usual scientific language. Perhaps say "As a first approximation we modeled …."

line 187 "..was estimated to drive…"

line 196 "As mention in the Introduction, "

line 201 ".…...caused by changes in gravitational separation." It is not clear whether you intend to say gravitational separation that occurred in the stratosphere, or gravitational separation that occurred in the troposphere. Please clarify. For example, "…caused by changes in stratospheric gravitational separation that influence the whole troposphere." OR "…caused by changes in gravitational settling within the troposphere itself."

line 261 "..seesaw.." perhaps you mean to say "inverse"?

line 265 " inputted " is an awkward word. Perhaps instead use "heat is added to a . . ."

line 269 " is non-negligible. . ."

line 275 "the derived. . ."

line 342 "…there is no method so far to validate OHC based on ocean temperature measurements." You will find a lot of oceanographers objecting to this statement. I would suggest you temper it somewhat, to something like "there is no method yet to adequately measure OHC via ocean temperature observations in the full-depth volume of the ocean".
* * *

---

## Author Comment (AC1) · 1 Oct 2020

**Responses to Referee 1:**

Thank you very much for your significant and useful comments on the paper "Secular change in atmospheric Ar/N$_2$ and its implications for ocean heat uptake and Brewer-Dobson circulation" by Ishidoya et al. We have revised the manuscript, considering your comments and suggestions. Details of our revision are as follows;

**Main Points**

1) **Surface observations. TKB has the longest and most dense data record. The observed temporal variations (Figure 2) give a compact annual cycle. In contrast, the sparser observations at the other 3 sites show much more variability (large amplitude variations). What are the reasons for this? Is it a measurement issue or possible real atmospheric variations?**

Lines 154-161: We consider the variability is a measurement issue. As you pointed out, the uncertainties of seasonal amplitude at COI, HAT and SYO are found to be larger than ±7 per meg of the uncertainty expected from the repeated analyses of the same flask air sample (Fig. 3). This would be due to the fact that the uncertainty of each analysis value of the standard air (±5.3 per meg, black dots in Fig. 1), which represents short-term (month-to-month timescale) stability of our δ(Ar/N$_2$) scale[*], is superimposed on the uncertainty of each analysis of the flask air sample and continuous measurement. Therefore, a mean squared error expected for the observational data from the flask air sample is about ±9 per meg, which is comparable to the uncertainties for the seasonal amplitudes in Fig. 3. Corrections of the δ(Ar/N$_2$) values at HAT and COI data prior to September, 2015 and January, 2019 (lines 94-105) could also be contributing to the uncertainties at the sites.

[*]It is noted that ±1.6 per meg is expected for the long-term (interannual timescale) stability of our δ(Ar/N$_2$) scale, as in Fig. 1.

2) **Observed long-term trend. Even the two longest data records (8 years) are short for deriving accurate long-term trends. The trend fit is not explained clearly. The model leaves variations which are > 36 months but the trend is quoted as if**

**it is a linear term? The fits to Ar/N2 and OHC in figure 2 are far from linear (but do vary with the long-term variability in the temperature data). I am concerned that the paper is reporting values in the abstract which imply a long-term linear trend, which is (quantitatively) not obvious from the plots.**

Lines 140-147, 216-232 and Fig. 5: The sentences and figure have been added to explain how the secular trend were obtained. We have added descriptions for the digital filtering technique and clarify the definition of the interannual variation and secular trend in the present study (Lines 140-147). We regard the average linear increasing/decreasing trend throughout the observation period as the secular trend. The methods used to extract the secular trends at TKB and HAT are described in Lines 216-232 and Fig. 5.

3) **Atmospheric modelling. An updated 2-D model has been used to model surface Ar/N2. There is no evaluation of the model in the stratosphere using available profiles of Ar/N2 to show that the higher altitude gravitational separation is modelled realistically.**

Lines 281-282, 407-420, Figs. A1 and A2: Considering your suggestions, the sentences and figures have been added to show the meridional distribution of the $\delta(Ar/N_2)_\Omega$ calculated using the updated SOCRATES model and its comparison with the stratospheric $\delta(Ar/N_2)$ over Japan and equatorial region observed in our previous studies.

4) **Imposed trend in the BDC. The stratospheric BDC is complicated with deep and shallow branches. A trend in the circulation is imposed in the model and the resulting trend at 35 km is shown. First, more information on the impact at other altitudes should be shown (e.g. latitude-height cross section of the impact on age-of-air). Second and more importantly, the use of the Engel et al paper to support a trend in AoA, which is converted to a correction of surface Ar/N2 is unjustified. Engel et al use a series of sparse balloon observations of CO2 and SF6 to derive an AoA trend (from 24-35km) up to 2005 – so there is no overlap with the observation period in this paper. Moreover, the error bar on their trend is very**

**large and the title of their paper gives the headline message of 'AoA unchanged within uncertainties'. Therefore, I cannot see how the imposed trend of 0.02 yrs/yr can be justified as the best estimate which gives the correction used in the abstract.**

Lines 293-304, 421-430 and Fig. A3: The sentences and figure have been added to discuss the annual mean meridional distribution of the AoA trend (yrs yr$^{-1}$) calculated using the updated SOCRATES model for weakened BDC simulation (lines 421-430 and Fig. A3), considering the first comment. For the second comment, we agree with you that the Engel et al. does not support the 0.02 yrs yr$^{-1}$ trend. In addition, we have recognized the simulations using the 2-D model in the present study, made by arbitrarily changing the MMC only with fixed horizontal mixing, are not enough to represent the mechanism to drive AoA change in the real atmosphere but it does served as a kind of sensitivity test. Therefore, we have clarified the limitation of the 2-D model and changed the reference from Engel et al. (2009) to Diallo et al. (2012) to support to use 0.02 yrs yr$^{-1}$ trend (lines 293-304). Of course, we know the data periods of Diallo et al. (2012) are not also overlap with the observational period in this study. However, we consider our simulation is worthwhile as a sensitivity test since it constitutes the first step to investigate the effect of gravitational separation of the whole atmosphere on the surface $\delta(Ar/N_2)$.

5) **Ocean model. This is a crude approach (as acknowledged by the authors) and it leads to statements in lines 187-193 that the model maybe too simplistic (i.e. not suitable) and that other factors may need to be considered. Overall, this part of the analysis seems incomplete therefore.**

Lines 196 and 205-215: As you pointed out, the one-box ocean model is not enough to evaluate responses of the atmospheric $\delta(Ar/N_2)$ to changes in the air-sea heat flux in detail. Unfortunately, we cannot use better ocean model to calculate spatiotemporal variations in the air-sea heat flux. Therefore, we have stated the limitation of the one-box ocean model clearer. We have also added a reference showing the renewal time of permanent pycnocline water in the North Pacific, which would make the readers imagine that the secular trend of the $\delta(Ar/N_2)$ for 8-years in the present study mainly reflects the OHC

change except deep ocean. We recognize the OHC changes estimated in the present study, based on the observed 8-years $\delta(Ar/N_2)$ trend combined with the 2-D atmospheric model and the one-box ocean model, are insufficient to suggest some revisions of the OHC from ocean temperature measurements. Nevertheless, it would of interest to see if it is possible to obtain a scientifically "meaningful" OHC change based on the firstly-reported secular $\delta(Ar/N_2)$ trend and the new concept of $\delta(Ar/N_2)_\Omega$.

**Other Specific Points**

1) **Line 21. The uncorrected trend of 0.75 +/- 0.30 per meg yr$^{-1}$ is also consistent with trend derived from ocean temperature, at the limits of the error bars. The correction is not needed, which is what is implied by the text.**

We leave the discussion as they are since the main aim of the correction is not an estimation of the precise OHC change comparable to that from ocean temperature measurements but to suggest that we cannot ignore the secular trend of $\delta(Ar/N_2)$ caused by changes in gravitational separation in the whole atmosphere.

2) **Line 132. 'Fig 2 in 3-1'. What does this mean?**

Lines 133-136: The sentence has been modified to make the meaning clearer.

3) **Line 163. 'Increase rates' (also in caption of Figure 4 and label axis). This should be referred to simply as 'rate of change'. The positive values will imply an increase.**

Lines 180-181 and Fig. 4: The words "increase rate" has been changed to "rate of change", as suggested.

4) **Line 180 'boldly'. This is the wrong word. You must mean something like 'crudely'.**

Line 197 and 381: The words "We boldly" has been changed to "As a first approximation we", considering your comments (line 197). The words "boldly assuming" has also been changed to "crudely assuming" (line 381).

**5) Line 183. 'modern' – better to say 'present-day'**

Line 200: The word "modern" has been changed to "present-day", as suggested.

**6) Line 184. Insert 'total atmospheric mass. . ..'.**

Line 201: The word "total mass" has been changed to "total atmospheric mass".

**7) Line 187 'drives'. Delete s.**

Line 204: The word "drives" has been changed to "drive". Thank you for pointing out.

**8) Line 216. 'molecular mass number' -> 'relative molecular mass'.**

Lines 254-255: The sentences have been modified. The $m_{A\text{-}air}$ is not a relative molecular mass, but a reciprocal average of $m_A$ and $m_{air}$.

**9) Line 221-223. Add a reference or make it clear that you are referring to this work.**

Line 264: We have added references.

**10) Line 238. 'decrease' (no s)**

Line 280: The word "decreases" has been changed to "decrease".

**11) Line 240 (and later). I think 'idealised' is better than 'virtual'.**

Line 283, 286 and 287: The word "virtual" has been changed to "idealised", as suggested.

**12) Line 250. Don't need to say 'increase'.**

Line 293: The word "increase rate" has been changed to "rate".

**13) Line 251. Give the dates that the Engel et al study covered (but see main comment above).**

Lines 299-300: We have added the dates that the Fritsche et al. (2016) covered and revised the sentence considering your main comment. Fritsche et al. is the study updated from Engel et al. (2009).

**14) Line 255. The word 'obtained' is wrong. The perturbation to the model circulation was forced arbitrarily. Change to e.g. 'forced'?**

Line 302: The word "obtained" has been changed to "forced", as suggested.

**15) Line 258. Nb 'simplistic' is a negative term which means that the model is too simple to be suitable.**

The sentence has been revised, and the word "simplistic" has not been used in the revised sentence.

**16) Lines 265. Change 'increases' to 'is estimated to increase'.**

Line 310: The words "atmospheric $\delta(Ar/N_2)$ increases" have been changed to "atmospheric $\delta(Ar/N_2)$ is estimated to increase".

**17) Line 267. Change 'expected' to 'is estimated'.**

Line 312: The word "expected" has been changed to "estimated".

**18) Line 269. Insert 'are non-negligible trends compared. . .'.**

Line 313: The words "are non-negligible compared" have been changed to "is a non-negligible trend compared".

**19) Line 274 Insert '. . . estimated effects..'.**

Line 317: The words "the effects" have been changed to "the estimated effects", as suggested.

**20) Caption for Fig. 2**

We have removed the incorrect sentence "All data are corrected for the scale drift of the primary standard air shown in Fig. 1 (b)." in the caption for Fig. 2 in ACPD paper since we have not applied such correction to the data.

---

## Author Comment (AC2) · 1 Oct 2020

**Responses to Referee 2:**

Thank you very much for your significant and useful comments on the paper "Secular change in atmospheric $Ar/N_2$ and its implications for ocean heat uptake and Brewer-Dobson circulation" by Ishidoya et al. We have revised the manuscript, considering your comments and suggestions. Details of our revision are as follows;

**The manuscript is challenging to review as it touches on a wide range of topics but information provided is often (too) sparse. As far as I can see, the main points of the manuscript are:**

**(i) Ar:N2 measurements are accurate enough to detect trends, which was not possible previously (Line 37).**

**(ii) Variations in the ratio over a decade are well correlated with OHC variations as reported by NOAA/NCEI.**

**(iii) There is a quantitative mis-match if only OHC is considered.**

**(iv) A model simulation with imposed change in the stratospheric BD circulation suggests that the near surface Ar:N2 ratio is as sensitivite to the BD circulation as to OHC variations (for the observed magnitude of changes in either of these).**

**(v) If one assumes that the OHC measurements and their impact on Ar:N2 are correct, one infers that over the period of measurements (2012-2019) the BD circulation slowed down.**

**The authors discuss the last point in some detail, and seem to be concerned that the sign of the change in the BD circulation is the opposite of a supposedly long term strengthening of the BD circulation. There is no reason, however, why a short period such as the one studied here should show the long term trend given the large interan nual variability of the stratospheric circulation. I recommend to shorten this somewhat misleading discussion (the authors are most likely not observing "trends"), and instead strengthen the points (i)-(iv).**

Thank you for your comments and suggestion. As you pointed out, the observational period of 8-years in the present study is not enough to discuss the long-term trend of the BDC. In addition, the troposphere, and not only the ocean, does not mix perfectly on a

timescale of a year, and that the surface-ocean temperature anomalies would be a large source of interannual variation on a yearly timescale for the observed $\delta(Ar/N_2)$ in the near-surface air. Therefore, we recognize the possibility that the secular trends observed at the surface and simulated by the 2-D model do not represent a long-term trend but is a part of the large interannual variation. Nevertheless, it would of interest to see if it is possible to obtain a scientifically "meaningful" OHC change based on the firstly-reported secular $\delta(Ar/N_2)$ trend at the surface and the new concept of $\delta(Ar/N_2)_\Omega$. Taking these into consideration, we have revised the related sentences throughout the manuscript, and stated clearly in the revised sentences that the 2-D model simulations are not enough to represent the mechanism to drive AoA change in the real atmosphere but it does serve as a kind of sensitivity test (the revised sentences have been highlighted by blue color : for example, lines 216-232, 293-304, and Appendix A and B).

**Specifically:**

**(a) This reviewer is not sufficiently familiar with the measurement technique to be able to assess claim (i); the data as presented in Figures 2,3 and 4 does look, however, reasonable; as does the calibration (Figure 1). Nonetheless, I hope that another reviewer may be in a position to comment.**

Thank you for your comments. As you expected, another reviewer commented on this matter and we have added the sentences not only to describe the uncertainties for seasonal cycles (lines 154-161) but also to show the detail method to extract the secular trend of $\delta(Ar/N_2)$ observed at the surface stations (lines 216-232).

**(b) The time filtering and frequency separation method needs to be better explained - based on the information provided in this manuscript, it is not possible to reconstruct what precisely the authors have done. This is an extremely important part of the argument as the claim is that seasonal variations are due to local SST variations, whereas the longer-term changes are due to global OHC changes (i.e. the seasonal cycle at TKB and HAT differ (Fig 3), but their longer-term variations (Fig 4) are highly correlated). Also, the analysis then uses largely**

**only TKB and HAT - but why not also show COI in Figure 4? Please use 2 different colors for AHT and TKB in Figure 4a - the separation with "+" and "o" does not allow to see the differences (Looking at Figure 2; it is surprising that these two timeseries align as well as shown in Figure 4).**

Lines 140-147 and Fig. 4: The sentences have been added to explain the time filtering and frequency separation method more in detail. Also, we have added the data at COI in Fig. 4 and used three different colors for TKT, HAT and COI, as suggested.

**(c) The SOCRATES model is not sufficiently described. Please provide more information on vertical and horizontal resolution, time steps, and parameterization of atmospheric transport and mixing. Please show the model's mean vertical profile of the Ar:N2 ratio, along with the measurements thereof in the stratosphere. Do you have other information (e.g. CO2 mixing ratio profile) that could serve as validation?**

Lines 245-246, 281-282, 295-296, 391-405, 407-430, Figs. A1, A2 and A3: Considering your suggestions, the sentences and figures have been added to show the detail descriptions of the SOCRATES model (lines 391-405), and meridional distribution of the $\delta(Ar/N_2)_\Omega$ calculated using the model and its comparison with the stratospheric $\delta(Ar/N_2)$ over Japan and equatorial region observed in our previous studies (lines 407-420, Figs. A1 and A2). We have also added the sentences and figure to discuss the annual mean meridional distribution of the AoA trend (yrs yr$^{-1}$) calculated using the SOCRATES model for weakened BDC simulation (lines 421-430 and Fig. A3). We have recognized the simulations using the 2-D model in the present study, made by arbitrarily changing the MMC only with fixed horizontal mixing, are not enough to represent the mechanism to drive AoA change in the real atmosphere but it does serve as a kind of sensitivity test. Therefore, we have clarified the limitation of the 2-D model (lines 293-304). We consider our simulation is worthwhile as the sensitivity test since it constitutes the first step to investigate the effect of gravitational separation of the whole atmosphere on the surface $\delta(Ar/N_2)$.

**Additional points**

1) **Figure 5: My understanding is that this is simply an idealized calculation for 10 years; if so, please change x-label (labels 2000 - 2010 suggest that this is specifically for this period; which is confusing also because OHC changes are taken from this period, see next item).**

Figure 6: Thank you for your comments, we have revised the figure as suggested. It is noted that the number of the figure has been changed from "Fig. 5" to "Fig. 6" since we have added a new figure as Fig. 5 to discuss the secular trends of the $\delta(Ar/N_2)$ observed at TKB and HAT.

2) **OHC: Why do you discuss OHC change for 2000-2010 (Line 267)? Why are you not using the data for 2012-2019? The text here is confusing; Figure 4b suggests you use the data for the correct period.**

Lines 310-312: The sentence has been modified to discuss OHC change for 2012-2019, considering your suggestion.

3) **Please indicate the baseline period for OHC (i.e. the figures shows departures from a baseline, not absolute OHC as one would think based on labels and caption).**

Lines 177-179 and caption of Fig. 4: The sentence and figure caption have been modified to show the baseline period for OHC, as suggested.

4) **Caption for Fig. 2**

We have removed the incorrect sentence "All data are corrected for the scale drift of the primary standard air shown in Fig. 1 (b)." in the caption for Fig. 2 in ACPD paper since we have not applied such correction to the data.

---

## Author Comment (AC3) · 1 Oct 2020

**Responses to Referee 3:**

Thank you very much for your significant and useful comments on the paper "Secular change in atmospheric $Ar/N_2$ and its implications for ocean heat uptake and Brewer-Dobson circulation" by Ishidoya et al. We have revised the manuscript, considering your comments and suggestions. Details of our revision are as follows;

**The one minor comment I have is that the one-box ocean model is somewhat over-emphasized, and perhaps the description could be simplified and shortened, because we know very well that the troposphere does not perfectly mix on a timescale of a year, nor does the ocean mix perfectly on a timescale of one year, so it is not really surprising that a one-box ocean model fails to match the observations of Ar/N2 at surface stations around Japan. I understand that the authors constructed the one-box model as a "straw man", to be shot down, but they could greatly simplify and shorten the discussion, while making it clear that they do not expect to be able to match their observations with a one-box model of the ocean. In fact, some readers may be confused, the way the authors have written about this one-box model (they seem to imply that they expected it to be able to match their observations). But of course the surface-ocean temperature anomalies are by far the largest source of noise on yearly timescales, for tracers like Ar/N2, measured in near-surface air. They should clarify that a true tropospheric- average Ar/N2, measured from aircraft (which is of course too expensive and so is prohibitive), would be needed to actually compare Ar/N2 observations with modeled Ar/N2. (So in some sense they are raising a false expectation with their one-box model exercise.)**
**On the whole, this is a groundbreaking and important paper, and will make a fine contribution to ACP. This is clearly excellent, highest-quality work!**

Lines 25-28, 196 and 205-215: Thank you very much for giving high evaluation to the present study. As you pointed out, the one-box ocean model is not enough to evaluate responses of the atmospheric $\delta(Ar/N_2)$ to changes in the air-sea heat flux in detail. Therefore, we have stated the limitation of the one-box ocean model clearer. We have also added a reference showing the renewal time of permanent pycnocline water in the

North Pacific, which would make the readers imagine that the secular trend of the $\delta(Ar/N_2)$ for 8-years in the present study mainly reflects the OHC change except deep ocean. We recognize the OHC changes estimated in the present study, based on the observed 8-years $\delta(Ar/N_2)$ trend combined with the 2-D atmospheric model and the one-box ocean model, are insufficient to suggest some revisions of the OHC from ocean temperature measurements. Nevertheless, it would of interest to see if it is possible to obtain a scientifically "meaningful" OHC change based on the firstly-reported secular $\delta(Ar/N_2)$ trend and the new concept of $\delta(Ar/N_2)_\Omega$.

**Minor editorial comments:**

**(1) line 35 ". . ..Argo floats".**

Line 39: The words "Argo float" have been changed to "Argo floats", as suggested.

**(2) line 47 - ". . .since it has been believed that the gravitational separation near the surface is too small to be detected..." This isn't really accurate, in the sense that there is no need for the gravitational settling process IN THE troposphere to be significant, in order for the stratosphere to affect the troposphere. Perhaps instead you could say something like, ". . .since it has been believed that the gravitational settling in the stratosphere is fairly small and constant in time, along with the fact that the troposphere has 10x more molecules than the stratosphere."**

**Or perhaps you meant to say, ". . .the gravitational separation signal from the stratosphere is too small to be detected at the surface."? This is accurate.**

Line 50-51: The words "since it has been believed that the gravitational separation near the surface is too small to be detected" have been changed to "since it has been believed that the gravitational separation signal from the stratosphere is too small to be detected at the surface", as suggested.

**(3) line 51 ". . .long-term changes in the Ar/N2 ratio near the surface are expected to be extremely small.."**

Line 54: The word "near the surface is" have been changed to "near the surface are".

**(4) line 53 again, it sounds like you are saying there there is gravitational separation near the surface, but this is not accurate. Maybe you mean the stratospheric gravitational separation signal near the surface?**

Lines 56-57: We have revised the phrase as "a very small secular change in the stratospheric gravitational separation signal near the surface may…", considering your comments.

**(5) line 58 ". . .secular trend of the Ar/N2 ratio. . ..."**

Line 62: The words "trend of $Ar/N_2$ ratio" have been changed to "trend of the $Ar/N_2$ ratio"

**(6) line 60 "Atmospheric Ar/N2 has been observed. . ..." (there is no need to include "ratio" here)**

Line 65: We removed the word "ratio", as suggested.

**(7) line 74 ". . ..and we usually use the average of 550 data values as the reported Ar/N2 ratio obtained from the continuous observations (about 11 hours of averaged data)."**

Lines 77-79: The sentence has been modified, considering your suggestion.

**(8) Line 101 ". . .glass flasks."**

Line 105: The words "glass flask" have been changed to "glass flasks".

**(9) Line 102 "per meg units as follows."**

Line 106: The words "per meg unit" have been changed to "per meg units".

**(10)     Line 112 ". . . corresponds to an uncertainty of. . ."**

Lines 115-116: The words "the uncertainty of" have been changed to "an uncertainty of".

**(11)     Line 116 ". . . to +1,800"**

Line 120: The word "1,800" has been changed to "+1,800".

**(12)     Line 122 "..but they did not correlate.."**

Line 125: The words "but did not" have been changed to "but they did not".

**(13)    Line 124 ". . . must have been superimposed. . ."**

Line 127: The words "must have superimposed on" have been changed to "must have been superimposed on".

**(14)    Line 126 ". . . due to a temperature. . ."**

Line 129: The words "due to temperature" have been changed to "due to a temperature".

**(15)    Line 138 ". . . by a fundamental sine-cosine. . ."**

Line 141: The word "fundamental" have been changed to "a fundamental sine-cosine".

**(16)    Line 143 ". . .reach seasonal maxima. . . due to the larger relative…" (enhancing is not really the correct word, sine the solubility is a physical constant and cannot be enhanced)**

Line 151: The word "enhancing" have been changed to "due to".

**(17)    line 146 "Similar increases. . .."**

Line 162: The word "increase" has been changed to "increases".

**(18)    line 150 "..found a seasonal. . ."**

Line 166: The words "found seasonal" have been changed to "found a seasonal".

**(19)    line 154 ". . .than the 14±6. . ."**

Line 170: The words "than 14±6" have been changed to "than the 14±6".

**(20)    line 160 this sentence is very long and hard to read. Perhaps you could simplify and shorten it. Also the verb "are shown" comes at the very end of the sentence, which is awkward. Instead you could write, "Variations in the 0-2000 m global OHC are shown (Fig. 4), reported by. . ."**

Lines 175-180: Considering your suggestion, we have revised the sentence as "Variations in the 0-2000 m global OHC are shown (Fig. 4), reported by the National Oceanographic Data Center (NOAA)/National Centers for Environmental Information (NCEI) (updated from Levitus et al. 2012, https://www.nodc.noaa.gov/OC5/3M_HEAT_CONTENT/). The OHC values are shown as anomalies from the baseline value observed in mid-1980s.

In the figure we also plot a interannual variation of the OHC values obtained by using the same digital filtering technique used in Fig. 2, and globally averaged surface temperature anomalies (Japan Meteorological Agency, http://www.data.jma.go.jp/cpdinfo/temp/nov_wld.html).".

**(21)     line 180 "We boldly modeled. . .." This is not usual scientific language. Perhaps say "As a first approximation we modeled . . ."**

Line 197: The words "We boldly modeled" have been changed to "As a first approximation we modeled", as suggested.

**(22)     line 187 "..was estimated to drive. . ."**

Line 204: The word "drives" has been changed to "drive". Thank you for pointing out.

**(23)     line 196 "As mention in the Introduction, "**

Line 235: The words "in Introduction" have been changed to "in the Introduction".

**(24)     line 201 ". . .caused by changes in gravitational separation." It is not clear whether you intend to say gravitational separation that occurred in the stratosphere, or gravitational separation that occurred in the troposphere. Please clarify. For example, ". . .caused by changes in stratospheric gravitational separation that influence the whole troposphere." OR ". . .caused by changes in gravitational settling within the troposphere itself."**

Lines 239-241: The sentence has been modified considering your suggestion as "Therefore, we need to explore the possibility of tropospheric $\delta(Ar/N_2)$ variations caused by changes in the stratospheric gravitational separation that influence whole troposphere.".

**(25)     line 261 "..seesaw.." perhaps you mean to say "inverse"?**

Line 306: The word "seesaw" has been changed to "inverse".

**(26)     line 265 "inputted" is an awkward word. Perhaps instead use "heat is added to a . . ."**

Line 310: The words "is inputted" have been changed to "is added to".

**(27)     line 269 " is non-negligible. . ."**

Line 313: The words "are non-negligible" have been changed to "is a non-negligible trend".

**(28)     line 275 "the derived. . ."**

Line 317: The word "derived" has been changed to "the derived".

**(29)     line 342 ". . .there is no method so far to validate OHC based on ocean temperature measurements." You will find a lot of oceanographers objecting to this statement. I would suggest you temper it somewhat, to something like "there is no method yet to adequately measure OHC via ocean temperature observations in the full-depth volume of the ocean".**

Lines 386-388: The sentence has been modified, as suggested.

**(30)     Caption for Fig. 2**

We have removed the incorrect sentence "All data are corrected for the scale drift of the primary standard air shown in Fig. 1 (b)." in the caption for Fig. 2 in ACPD paper since we have not applied such correction to the data.

---

## Author Response (AR2)

**Responses to Referees 1 and 2:**

Thank you very much for your significant and useful comments on the paper "Secular change in atmospheric $Ar/N_2$ and its implications for ocean heat uptake and Brewer-Dobson circulation" by Ishidoya et al. We have revised the manuscript, considering your comments and suggestions. Details of our revision are as follows;

*Comments from Referee 1*

1) **Abstract. Lines 18-25. I still find the discussion about the need for the modelled correction to the $Ar/N_2$ trend and the consistency with the ocean temperature measurements misleading. Basically, the modelled correction to $\delta(Ar/N_2)$ is about half of the observational uncertainty (0.15 v 0.30 per meg yr$^{-1}$). The best estimates from the uncorrected and corrected trends (0.75, 0.60, 0.88 per meg yr$^{-1}$) are \*all\* much larger than that from the ocean temperature data. However, the error bars are large and so both the uncorrected and weakened-BDC trend are consistent with the temperature data. Any correction for a weakening trend will push the large overestimate in the right direction. A correction in the other direction (stronger BDC) will push the trend in the wrong direction and outside of the range of even the large error bars. I think that the abstract is too strong on suggesting that this BDC correction is key to getting agreement.**

Lines 19-26: We have revised the related sentences in the abstract as follows, considering your comments.

"The secular trend of the $Ar/N_2$ ratio at TKB, corrected for gravitational separation under the assumption of weakening (enhancement) of BDC simulated by the 2D model, was $0.60\pm0.30$ ($0.88\pm0.30$) per meg yr$^{-1}$. By using a conversion factor of $3.5\times10^{-23}$ per meg J$^{-1}$ by assuming a 1-box ocean with a temperature of 3.5 °C, average OHC increase rates of $17.1\pm8.6$ ZJ yr$^{-1}$ and $25.1\pm8.6$ ZJ yr$^{-1}$ for the period 2012 – 2019 were estimated from the corrected secular trends of the $Ar/N_2$ ratio for the weakened and enhanced BDC conditions, respectively. Both the OHC increase rates from the uncorrected and weakened-BDC secular trends of the $Ar/N_2$ ratio are consistent with $12.2\pm1.2$ ZJ yr$^{-1}$ reported by ocean temperature measurements, while that from the enhanced-BDC is outside of the range of the uncertainties."

**2) Page 6. Line 184. 'long-term change'. The strong correlation applies to the changes over a few years, not the long-term change. It is an assumption that the correlation will also apply to small long-term trends.**

Lines 185: The words "long-term change" has been changed to "interannual variations", considering your suggestion.

*Comments from Referee 2*

**1) The authors have addressed most of my suggestions and concerns. The paper is not overstating the results, and is a very valuable contribution. The main result is that δ(Ar/N₂) is sensitive to both OHC changes and BDC changes - without a clear dominance of one of the processes. The authors then show that the measurements are more conistent with a slow-down of the BDC over the short period considered here. I recommend minor revisions: (i) Generally, proof-reading by a native speaker would help (maybe Copernicus offers such a service?) (ii) The abstract should be more precise what "weakened" and "enhanced" BDC mean in this study - it is peculiar to read sensitivity results with error bars, but no quantification of the perturbation is provided. (iii) Around Line 243: The SOCRATES model is introduced without any explanation, rather the reader is pointed to the Appendix. I would like to see a minimal description in the main text: It's a 2D model with parameterized eddy diffusivity coefficients, tuned to produce a realistic stratospheric age of air distribution (I'd assume - but this is the information I'd like to read here). Then - a sentence along the line of "Sensitivity studies are carried out with arbitrarily modifying the mass stream function in the model." (i.e. what is now only in the Appendix around Line 422.)**

Lines 16-19: We have revised the related sentence in the abstract as follows, considering your comment (ii).

"In order to examine the possibility of the secular trend in the surface Ar/N₂ ratio being modified significantly by the gravitational separation in the stratosphere, 2-dimensional model simulations were carried out by arbitrarily modifying the mass stream function in

the model to simulate weakening an enhancement of the Brewer-Dobson circulation (BDC)."

Lines 244-248: Short descriptions of the SOCRATES model have been added as follows, considering your comment (iii).

"Therefore, in this study we updated the SOCRATES model (Huang et al., 1998) to calculate variations in Ar and $N_2$ from the surface to 120 km, taking into account molecular diffusion processes generating gravitational separation. The SOCRATES is a 2-D model with parameterized eddy diffusivity coefficients, tuned to produce a realistic stratospheric age of air distribution. In the present study we carry out sensitivity studies with arbitrarily modifying the mass stream function in the model."

As for your comment (i), the manuscripts, published in ACPD and its revised version, have been proofread by a native speaker. Therefore, we have not requested the native speaker to proofread our manuscript again.

[revised manuscript text omitted]